# Scaling crossover in droplet impact force on elastic substrates

Yuto Yokoyama [1,2], Hirokazu Maruoka[3], Kaie Matsunuma[2] & Yoshiyuki Tagawa [2,4] ✉

Droplet impacts are fundamental to fluid-structure interactions, shaping processes from erosion to bioprinting. While previous scaling laws have provided insights into droplet dynamics, force scaling laws remain insufficiently understood, particularly for soft substrates where both the droplet and substrate deform significantly. Here, we show that droplet impacts on elastic substrates exhibit a scaling crossover in maximum impact force, transitioning from inertial force scaling, typical for rigid substrates under high inertia, to Hertzian impact scaling, characteristic of rigid spheres on elastic substrates. Using high-speed photoelastic tomography, we captured high-resolution dynamic stress fields and identified a similarity parameter governing the interplay between droplet inertia, substrate elasticity, and deformation time scales. Our findings redefine how substrate properties influence impact forces, demonstrating that droplets under high inertia-long thought to follow inertial force scaling-can instead follow Hertzian impact scaling on soft substrates. This framework provides practical insights for designing soft, impact-resistant materials.

The behavior of droplets impacting solid surfaces affects the quality and efficiency of many industrial processes, such as inkjet printing[1] and spray cooling[2,3]. The ubiquity and importance of droplet impact make it one of the most extensive areas of research in the fluid mechanics field[4,5] over a period of about 150 years, starting since Worthington's pioneering work[6,7]. While research has frequently concentrated on the morphology of droplets at the moment of impact[8,9], it has been demonstrated that scaling laws are crucial in explaining these complex phenomena. In particular, scaling laws for droplet contact time[10,11] and maximum spreading diameter[12–14] have provided valuable insights into droplet dynamics. These morphology-based scaling laws have provided the foundation for understanding a wide range of droplet behaviors on rigid substrates.

However, as noted by Cheng et al.[15], relatively few studies have examined the forces and stress distributions generated during droplet impact, while most work has focused instead on post-impact droplet morphology as mentioned above. In fact, the impact force and stress

produced by droplet impact are essential in various industrial technologies, such as turbine blade erosion by raindrops[16,17], cutting techniques such as water cutters[18,19], and next-generation printing, such as bioprinting[20,21]. In particular, the maximum impact force is a key parameter in many impact-related applications, and its accurate prediction is essential.

Given its industrial relevance, recent studies have extensively investigated the factors controlling the maximum impact force, $F_{max}$[22–25]. We briefly summarize the key findings of these investigations below. Soto et al.[26] experimentally demonstrated that $F_{max}$ is proportional to the inertial force of the droplet in the high-Reynolds-number regime (Re = $\rho V R/\eta$, where $\rho$, $\eta$, $V$, and $R$ are the density, viscosity, impact velocity, and initial radius of the impacting droplet, respectively). Specifically, they found that

$$F_{max} \propto \rho V^2 R^2. \qquad (1)$$

[1]Micro/Bio/Nanofluidics Unit, Okinawa Institute of Science and Technology Graduate University, Onna, Okinawa, Japan. [2]Department of Mechanical Systems Engineering, Tokyo University of Agriculture and Technology, Koganei, Tokyo, Japan. [3]Nonlinear and Non-equilibrium Physics Unit, Okinawa Institute of Science and Technology Graduate University, Onna, Okinawa, Japan. [4]Institute of Global Innovation Research, Tokyo University of Agriculture and Technology, Koganei, Tokyo, Japan. ✉e-mail: tagawayo@cc.tuat.ac.jp

This indicates that the dimensionless maximum impact force, $\tilde{F}_{\max} = F_{\max}/(\rho V^2 R^2)$, is constant in the high-Re limit, which is called "inertial force scaling" in the review by Cheng et al.[15]. Zhang et al.[27] examined the impact force of water droplets on superhydrophobic surfaces. They discovered that a second force peak occurs due to cavity collapse inside the droplet during the retraction phase, following the first peak (which has also been measured on hydrophilic substrates) and the maximum spreading of droplets. Cavity collapse typically occurs on the non-wetting surface at moderate Weber numbers (We = $\rho V^2 R/\gamma$, where $\gamma$ is the surface tension of the liquid) and the relatively low Ohnesorge numbers (Oh = $\eta/\sqrt{\rho R\gamma}$)[28,29]. Sanjay et al.[30] find that, on non-wetting surfaces at low Ohnesorge numbers (Oh = $\lesssim 0.1$) and at moderate Weber numbers (We $\lesssim 10$), the second peak can exceed twice the inertial contribution to the impact force ($\rho V^2 R^2$), while increased viscosity can suppress the second peak. Sanjay et al.[31] reported that substrate wettability (contact angle) does not influence the first peak of a water droplet, while it does influence the second peak. They showed that greater wettability (low contact angle) reduces, and in some cases eliminates, the second peak in the impact force, which becomes indistinguishable for contact angles below about 40°. Furthermore, at intermediate impact velocities, the water droplets bounce in a ring shape, producing a third impact force peak when hydrophobic beads are placed below the droplet[32]. On the other hand, when the droplet viscosity increases (low-Re regime), the spreading becomes smaller while the impact force is significant, i.e., $F_{\max}$ increases as droplet viscosity decreases[23,33]. These observations indicate that the rheological properties of droplets, in other words, droplet deformability (how fast the droplet can spread), influence $F_{\max}$. The variation of $F_{\max}$ with respect to Re was recently formulated theoretically[14,30,33]. However, in the work by Gordillo et al.[33], the data for $F_{\max}$ deviate from the predictions at Re $\lesssim 1$, and they mentioned that Re is no longer a suitable dimensionless number for scaling $\tilde{F}_{\max}$ in this region.

These findings imply that a soft substrate can alter droplet deformability and thus influence $F_{\max}$. When the substrate is sufficiently soft, substrate deformation may suppress droplet deformation. This effect is expected to be more pronounced as the droplet viscosity increases. In this situation, $F_{\max}$ is expected to exhibit "Hertzian impact scaling", i.e.

$$F_{\max} \propto \rho^{3/5} V^{6/5} R^2 E^{2/5}, \tag{2}$$

as is observed when a rigid sphere impacts an elastic substrate with an elastic modulus of $E$[34–36]. However, the transition process of scaling law has never been observed, and the effect of the substrate elasticity on $F_{\max}$ is not clarified.

In parallel, as reviewed by Mohammad Karim[37], many studies have addressed the morphology of droplets impacting onto flexible substrates. Mangili et al.[38] and Alizadeh et al.[39] suggested that energy loss occurs due to the moving of the wetting ridge, which is the soft substrate deformation due to the surface tension at the three-phase contact line around the droplet[40,41], and it decreases the droplet retraction velocity after maximum spreading. A softer substrate suppresses splashing by decreasing the pressure inside the droplet near the contact line when splashing[42] and reduces the maximum spreading diameter caused by the energy dissipation resulting from substrate deformation[43]. However, despite the importance of forces and stresses generated during droplet impacts, their dynamics remain less explored experimentally for soft substrates.

Investigating the stress field in the substrate, which changes in spatial distribution depending on the impacting droplet behavior, is also crucial for understanding such a phenomenon involving fluid-structure interaction. Nonetheless, experimental measurement of the stress field in the substrate during droplet impact was not achieved until very recently due to the requirement for high spatio-temporal resolution. Sun et al.[44] were the first to successfully measure the stress field in the elastic substrate during droplet impact using digital image correlation. However, their experimental conditions were limited, and the effect of the substrate elasticity on the maximum impact force remains unclear. Their contribution has enabled quantitative investigation of the stress field, opening up new possibilities for research on such a dynamic fluid-structure interaction problem. However, to understand the impact force and stress in a wider parameter space, it is necessary to develop new stress measurement methods in addition to the above technique, as indicated in the latest review[15].

Here, we applied our novel optical stress measurement technology called "high-speed photoelastic tomography"[36] to quantify the stress field in elastic substrates during droplet impact. By varying droplet viscosity and substrate elasticity, we demonstrate for the first time that as the droplet's viscosity increases, the scaling law of the maximum impact force $F_{\max}$ shows a crossover from inertial force scaling, Eq. (1), to Hertzian impact scaling, Eq. (2), based on the stress field visualization. To explain the physics behind this crossover and how the substrate elasticity $E$ is involved in the process, we also employ a data-driven algorithm to identify similarity parameters bridging the two asymptotic regimes.

## Results and Discussion
### Dynamic stress field and impact force
In the experiment, we recorded the behavior of the impactor (a rigid sphere or silicone oil droplets with different viscosities) and the six transparent elastic substrates (polyurethane and gelatin gels with different elastic moduli) by using the high-speed polarization camera. The polarization camera can access the polarization state, which is called photoelastic parameters, of the light passing through the stressed substrate. The polarization state is used for the stress reconstruction based on photoelastic tomography (see Methods for details). Figure 1 shows the sequence of grayscale images of the impactor (top), the photoelastic parameters (retardation $\Delta$ and orientation $\phi$, bottom left), and the reconstructed axial stress ($\sigma_{zz}$, bottom right) in the substrate (Gel III) with a moderate elastic modulus of 47.4 kPa. The time at which the impactor touches the substrate is defined as $t = 0$ s. Gray-scale images show that, after impact, the highly viscous droplets do not spread much due to the viscous dissipation[45], whereas the low-viscosity droplets spread significantly. When impacted by the rigid sphere, the substrate deforms up to approximately 40% of the sphere radius. The substrate deformation during droplet impact is so slight that it cannot be clearly observed with the current spatial resolution of the image.

The retardation $\Delta$, which corresponds to the integration of secondary principal stress difference within the substrate[46,47], is larger for sphere impact than for droplet impact, as shown by the difference in the color bar in Fig. 1. This means that the stress induced by the sphere impact is much larger than that of the droplet impact. In the case of droplets, the maximum retardation, i.e., maximum stress, increases with the droplet viscosity. The orientation $\phi$ (indicated by white arrows), which is related to the direction of secondary principal stress within the substrate, is directed vertically overall in the area below the droplet-substrate contact. In response to this, it is directed horizontally in the outer region of the area. This behavior means that the substrate is deforming as an elastic half-space rather than as a simple Winkler elastic foundation, which does not have the interaction between the adjacent elements in a horizontal direction[34].

In the bottom-left panels of Fig. 1, along the $r$-axis of $\sigma_{zz}$ at $t \simeq 0.1$ ms, in the region near the contact area, $\sigma_{zz}$ is positive, whereas in the outer region, $\sigma_{zz}$ is negative. Positive values of $\sigma_{zz}$ indicate that the substrate is compressed in the $z$-axis, and negative values indicate tensile deformation in the $z$-axis. Immediately after impact, the boundary between positive and negative stresses follows the rim of the contact area. It then propagates in the $r$-direction beyond the contact

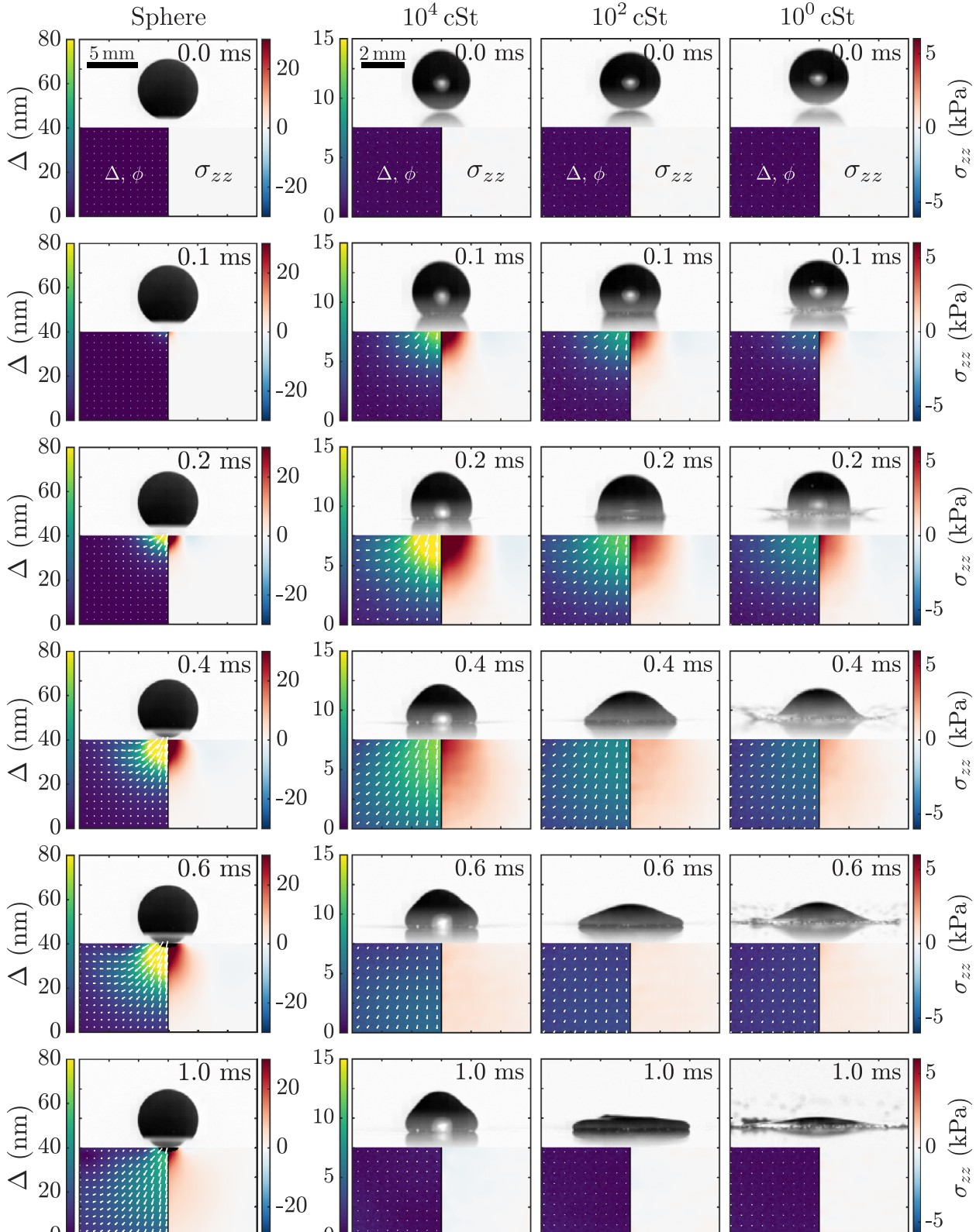

**Fig. 1 | Dynamic behaviors of the impactors and the substrate response.** Spatio-temporal distribution of the stress field (bottom-left panels) photoelastic parameters, $\Delta$ and $\phi$, and (bottom-right panels) reconstructed axial stress $\sigma_{zz}$ when a sphere and droplet impact on the elastic substrate, Gel III ($E = 47.4 \,\text{kPa}$) with $V \simeq 2.8 \pm 0.1 \,\text{m/s}$. In the bottom-left panels, the colormap indicates the retardation $\Delta$ and white arrows indicate the orientation $\phi$. The arrow length corresponds to the magnitude of retardation shown in the colorbars. The gray-scale images of the impactor on the upper panel are obtained by subtracting the background image without the impactor.

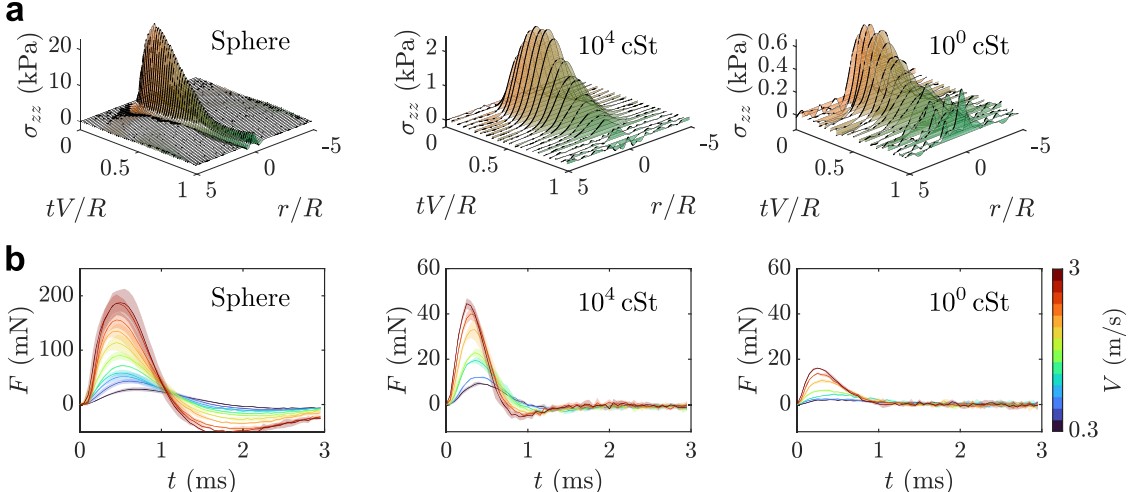

**Fig. 2 | Stress and force acting on the elastic substrate. a** Temporal evolution of the axial stress distribution $\sigma_{zz}$ on the elastic substrate (Gel III, $E = 47.4\ kPa$) at $z = 0$, when Cauchy number, Ca = $\rho V^2/E$, is approximately 0.02. The data were smoothed by MATLAB function "`smoothdata`". **b** Temporal evolution of the impact force with different impact velocities during a sphere or droplet impact on the elastic substrate, Gel III ($E = 47.4\ kPa$). The shaded regions represent one standard deviation for three experiments.

area. The maximum stress of $\sigma_{zz}$ appears approximately 0.2 ms after impact, and its value increases with increasing droplet viscosity. At this point, the $10^0$ cSt silicone oil droplet spreads into a hemispherical shape and exhibits splashing at high impact velocities. On the other hand, the spreading of the $10^4$ cSt silicone oil droplet is suppressed by viscous dissipation, maintaining a shape closer to a sphere. These differences in droplet dynamics during impact are reflected in the magnitude of the generated stress. At $t = 1.0\ ms$, the highly viscous droplets, e.g., $10^4$ cSt silicone oil, produce negative values of $\sigma_{zz}$ even below the droplet–substrate contact area. This is because the substrate, which has been slightly deformed downwards by droplet impact, deforms upwards to return to its original position due to its elasticity. This negative stress is more pronounced in the sphere case at later times (see Fig. 2 of the Supplementary Information) and is barely noticeable in low-viscosity droplets, which produce little deformation of the substrate.

Figure 2a shows the change in axial stress distribution $\sigma_{zz}$ at $z = 0$ over time. In the case of the sphere, the stress region near the center remains narrow because the sphere cannot spread. The $10^4$ cSt silicone oil droplet exhibits a similar behavior due to its high viscosity, which prevents rapid spreading of the axial stress along the $r$-axis. In contrast, the $10^0$ cSt silicone oil droplet shows a wider propagation of stress distribution along the $r$-axis because of the faster spreading of the droplet. However, it does not show the non-central stress peak, which is predicted by many theoretical and numerical works[17,42,48–50] and experimentally observed by Sun et al.[44]. They indicated that if the substrate is elastic, the peak position of $\sigma_{zz}$ becomes closer to the center, $r = 0$, which is not expected if the substrate is infinitely rigid. Based on the findings of Sun et al.[44], we hypothesized that the non-central peak was not observed as distinct because the substrate used in this study was softer than theirs (minimum 100 kPa), causing it to may merge with the stress field near the center. The experimental data, obtained using substrates with different elastic moduli, also do not show the distinct non-central peak (see Fig. 4 in the Supplementary Information). However, it cannot be ruled out that our observations may be due to experimental artifacts. In particular, it has been reported that the reconstructed $\sigma_{zz}$ overestimates the actual value near the $z$-axis[51], which is an inherent property of the axisymmetric field reconstruction algorithm based on the Abel inversion ($r = 0$ is a singular point, and noise accumulates toward the center). In fact, the reconstructed stress field shows relatively large peaks near $r = 0$ as

shown in Fig. 4 in the Supplementary Information. This suggests that the stress field that should appear lower inside the non-central peak may not be captured. To confirm whether the non-central peak shifts toward the center in a softer substrate, it is necessary to perform further experiments with a higher spatio-temporal resolution setup using a reconstruction algorithm that avoids errors near the symmetry axis. On the softest substrate, we also observe a pronounced tensile (negative) stress region localized near the liquid-solid contact line (see Fig. 5a in the Supplementary Information). This feature is also present in the rigid sphere impact case and high-viscosity droplet ($10^4$ cSt silicone oil) case (see Fig. 1 and Fig. 2 Supplementary Information).

Figure 2b exhibits the temporal evolution of impact force, $F(t)$, acting on the substrate Gel III ($E = 47.4\ kPa$). The impact force $F(t)$ increases rapidly and reaches its maximum within 1 ms. After showing the maximum impact force $F_{max}$, $F(t)$ decreases and reaches zero in later times. The sphere and $10^4$ cSt silicone oil droplet show the negative force at $t \sim 1\ ms$ due to the substrate deformation induced by the impact. Previous studies have reported that when the substrate is superhydrophobic, a second peak in impact force appears due to the retraction behavior of the droplet[25,27]. For each droplet impact, the maximum value of the impact force, $F_{max}$, increases with increasing impact velocity $V$ (i.e., droplet inertia) and droplet viscosity. $F_{max}$ is roughly ranging from 0.9 to 47 mN for the droplet case with varying impact velocity. Comparing cases with similar impact velocities, the sphere exhibits much higher $F_{max}$ than droplets. $F_{max}$ of sphere reaches approximately 220 mN. For the sphere case, while $t \lesssim 1\ ms$, $F(t)$ is symmetric with respect to the time when $F_{max}$ appears if $V$ is sufficiently high. This is expected under Hertzian impact theory[34]. $F(t)$ is also symmetric for the high-viscosity droplets while $t \lesssim 0.7\ ms$. However, for low-viscosity droplets, $F(t)$ exhibits an asymmetric shape while $t \lesssim 1\ ms$. These tendencies are similar to the results of previous studies[23,33], although this earlier work used non-deformable rigid substrates, whereas the present experiment uses deformable elastic substrates. Notably, for the softest substrate (Gel VI), the temporal profile $F(t)$ becomes more symmetric about the time of $F_{max}$ (see Fig. 5b in the Supplementary Information) than on stiffer substrates, such as Gel III, which is consistent with a trend toward a Hertzian-like response.

In our experiments, the impact-force traces are single-peaked, i.e., no secondary maximum is observed, unlike reports on non-wetting substrates[27,30,31]. Under comparable conditions within the same fluid family (silicone oils) and at high Weber numbers

(We = $\rho V^2 R/\gamma \approx 100–500$), Gordillo et al.[33] likewise measured a single peak followed by a monotonic decay, consistent with our data (see Fig. 6 in the Supplementary Information for our parameter space). When it appears, the second force peak originates from cavity collapse after maximum spread during retraction[4,29]. Its prominence increases on non-wetting surfaces and depends explicitly on contact angle[25,27,31,32]. Zhang et al.[31] reported that the second force peak is strongly suppressed or absent for sufficiently wettable substrates (contact angle is less than approximately 40°). Because our substrates are wettable by silicone oils, we did not observe the retraction dynamics that generate an internal cavity, nor any subsequent cavity collapse. The missing second peak therefore reflects the physics of our regime rather than a limitation of the force/stress measurements. Direct, time-resolved mapping of the substrate stress at the instant of cavity collapse remains an important direction for future work.

Sudden liquid deceleration during impact can generate the water-hammer pressure of order $\rho V c$, where $c$ is the speed of sound within the droplet, acting over a time duration $RV/c^2$ on an area $(RV/c)^2$. It yields a force scale $F_w \propto \rho R^2 V^3/c$[15,26,52,53]. We confirmed that the ratio $F_{max}/F_w$ is significantly larger than 1 in our experiment (see Fig. 7 in the Supplementary Information), indicating that the water-hammer effect can be negligible.

Air cushioning, in which an air layer of 0.1-10 $\mu m$ is trapped between the droplet and the substrate in the first nanoseconds to microseconds after impact[54,55], further weakens the water hammer pressure reported by Hoksbergen et al.[17]. Moreover, Langley et al.[56] experimentally revealed that a softer substrate can trap a larger air layer than a harder substrate (implying an even smaller $F_w$ in our elastic substrates). The droplet viscosity also affects the air cushioning[57]. While the effect of air cushioning is believed to be minimal during the maximum impact force (at hundreds of microseconds) of interest in this study, it may affect the time scaling of $F(t)$ and the non-central peak of the stress field. However, as noted by Cheng et al.[15], there is a lack of systematic experimental investigations into the effects of air cushioning on the force and stress of impacting droplets. Combining our stress measurement system with high-speed interferometry[58] will enable the simultaneous tracking of air-layer evolution and substrate stresses, clarifying how droplet viscosity and substrate elasticity influence this fluid-structure interaction. In our regime, direct AFM scans of the soft gel were unreliable; instead, the acrylic mold used to cast the gel blocks exhibited a surface roughness of around 10-100 nm (see Fig. 8 in the Supplementary Information). The gel surface is expected to be slightly rougher than the mold. The characteristic air-film thickness at early times is $\sim 0.1\,\mu m$, and prior studies report micron/sub-micron gas films that collapse on microsecond time

scales[59–62]. Because the estimated roughness is comparable to the minimum gas-film thickness, the air layer should rupture and direct liquid-solid contact be established before 100 μs (the time of our measured force peak). Thus, air cushioning is expected to play a negligible role in our measured forces and stresses. On smoother surfaces, however, the air layer can persist and redistribute the pressure field (e.g. screen the water-hammer pressure and shift the pressure maximum off-axis)[15,17,63]. Comparative measurements under reduced ambient pressure would be decisive for isolating air-cushion effects, and we identify them as important future work. Finally, the absence of a second force peak in our data is unlikely to originate from the initial air layer; rather, it reflects later-time hydrodynamic events in our parameter regime. Although crucial for various processes, stress fields in the very earliest instants after impact (first few microseconds) are not resolved here due to measurement limitations.

Wetting ridge may be formed due to the softness of the substrate in our experiment. However, its height and the force generated are predicted to be much smaller than the spatial resolution of our optical setup and the minimum $F_{max}$ measured in our experiment, respectively (see Supplementary Information for details). Therefore, the contribution of the wetting ridge to impact force and stress is not considered in this study.

## Crossover scaling in impact forces

Here, we discuss the scaling of our problem, i.e., how the maximum impact force is influenced by the physical quantities, especially the droplet viscosity and the substrate elasticity. The physical parameters of interest are $F_{max}, \rho, V, R, \eta$, and $E$. Let us assume that $F_{max}$ is described by a function, $F_{max} = g(\rho, V, R, \eta, E)$. As noted above, we assumed that the effect of the water-hammer pressure can be ignored in this study. We introduce the following nondimensionalization. By selecting $\rho, E, R$ as physical parameters with independent dimensions[64], we naturally obtain the relationship between the dimensionless numbers, $\Pi = f(\theta, Ca)$, where $\Pi = F_{max}/(ER^2)$, $Ca = \rho V^2/E$ (the Cauchy number), and $\theta = \eta/\sqrt{\rho ER^2}$. The relationships between them and the dimensionless numbers $\tilde{F}_{max}$ and Re are $\Pi = \tilde{F}_{max} Ca$ and $\theta = \sqrt{Ca}/Re$.

The scaling relations between $\Pi$ and $Ca$, incorporating data from six substrates of different elasticity, are shown in Fig. 3a. They are illustrating the different power-law behaviors depending on $\theta$, i.e., droplet viscosity, and two distinct behaviors can be found. The first is Hertzian impact scaling[34], where $\Pi \propto Ca^{3/5}$ for large $\theta$; see also Eq. (2). The second is inertial force scaling[15], where $\Pi \propto Ca$ for small $\theta$; see also Eq. (1). The slope for the $10^4$ cSt silicone oil droplet is close to 3/5, consistent with Hertzian impact scaling. As droplet viscosity

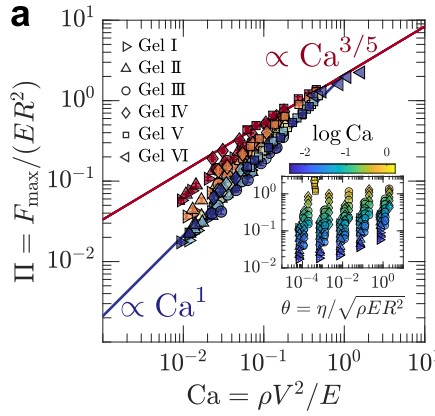
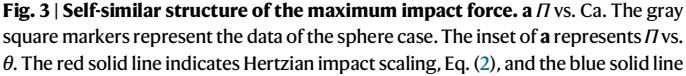
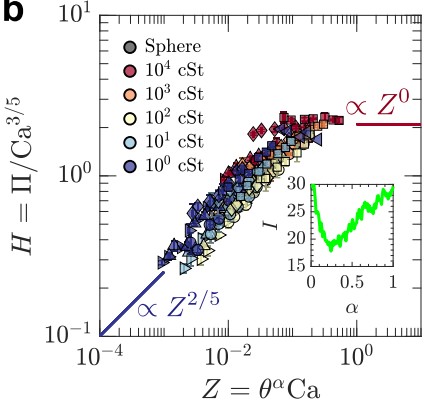

**Fig. 3 | Self-similar structure of the maximum impact force. a** $\Pi$ vs. Ca. The gray square markers represent the data of the sphere case. The inset of **a** represents $\Pi$ vs. $\theta$. The red solid line indicates Hertzian impact scaling, Eq. (2), and the blue solid line indicates inertial force scaling, Eq. (1). **b** $H$ vs. $Z = \theta^\alpha$Ca with $\alpha = 0.24$. The red solid line indicates Eq. (3) and the blue solid line indicates Eq. (4). The inset of **b** shows $I$ vs. $\alpha$. The error bars represent one standard deviation over three impacts.

decreases, the slope increases toward 1, as in inertial force scaling. Moreover, substrates of different elasticity exhibit similar slope behavior for the same droplet viscosity (see Fig. 9 in the Supplementary Information). This interpretation is a consistent picture with the observation on the stress and force shown in Figs. 1 and 2. An exception to this overall trend appears on the softest substrate (Gel VI, $E \approx 2.7$ kPa). Even for a low-viscosity droplet (1 cSt), the slope in the Π–Ca plot aligns with the trend of Hertzian impact scaling. This suggests that sufficiently soft substrates can enhance Hertzian-like behavior, even when the liquid viscosity is low. The Gel VI force signature (Fig. 5b in the Supplementary Information) as described before also primarily supports this interpretation. At the same time, however, the pronounced negative stress near the contact line and the irregular damping in $F(t)$ are suggestive of an increasing role of substrate viscosity (viscoelasticity) in the ultra-soft limit, implying that our present set of similarity parameters may be incomplete in that regime. A more decisive assessment would require further softening of $E$, higher impact speeds, or systematic viscosity sweeps with gels that are both very soft and sufficiently photoelastic for repeatable measurements. Furthermore, as the plots do not collapse under these dimensionless numbers, the problem is likely to fall into the self-similarity of the second kind, for which the self-similarity cannot be captured solely through dimensional analysis[64] (see Methods for the detailed explanation about the self-similarity).

As the scaling behavior changes depending on $\theta$, including $\eta$, $\theta$ is a driving parameter that governs the crossover of scaling law. In the limit of larger $\theta$ (or larger Ca), the scaling follows the Hertzian impact scaling. Therefore, the function, $f$, must satisfy the following behavior in this limit, $f(\theta, \mathrm{Ca}) \propto \mathrm{Ca}^{3/5}$. To consider the self-similar solution that governs the crossover, we define the following dimensionless number and the function, $H = \Pi/\mathrm{Ca}^{3/5}$ and $\Phi(\theta, \mathrm{Ca}) = f(\theta, \mathrm{Ca})/\mathrm{Ca}^{3/5}$, according to the framework of Maruoka[65]. In the Hertzian impact scaling as large $\theta$, $\Phi$ must satisfy

$$\Phi(\theta, \mathrm{Ca}) \propto \mathrm{const.} (\theta \gg 1). \tag{3}$$

On the other hand, the inertia force scaling ($\Pi \propto \mathrm{Ca}$) is recovered in small $\theta$. Thus $\Phi$ follows

$$\Phi(\theta, \mathrm{Ca}) \propto \mathrm{Ca}^{2/5} (\theta \ll 1). \tag{4}$$

$\Phi$ should satisfy these two asymptotic behaviors while $\Phi$ is still indeterminate. However, assuming that $\Phi$ belongs to the similarity of the second kind, $\Phi$ has the potential to have the following form: $\Phi(\theta, \mathrm{Ca}) = \Phi(\theta^\alpha \mathrm{Ca})$. Using the prayer beads algorithm (see Methods) in which the power exponent of the similarity parameter, $\alpha$, was optimized so that the sum of the distance of each data plots $I(\alpha)$ is minimized, $\alpha$ was estimated as $\alpha \simeq 0.24$ while the minimized values are located around 0.1–0.5, as shown in the inset of Fig. 3b. Plotting the data points using the similarity parameters $H$ and $Z = \theta^\alpha \mathrm{Ca}$ as Fig. 3b, the plot reveals seemingly reasonable data collapse, and the asymptotic behavior follows the condition for the crossover of scaling laws, i.e., Eqs. (3) and (4). Notably, for the $10^4$ cSt droplet on the substrate Gel V ($E = 17.9$ kPa) and for the 1 cSt droplet on the substrate Gel VI ($E = 2.7$ kPa), $H$ is approximately constant with $Z$ (i.e., $H \propto Z^0$). From the $H$ versus $Z$ data, we propose a critical value $Z_c \approx 1$ for the transition between the two scaling laws when $\alpha = 0.24$. By calculating $Z$ from the experimental parameters, $Z_c$ can then be used to predict whether the impact exhibits behavior closer to one of the scaling laws, depending on whether it is larger or smaller than $Z_c$.

At present, the theoretical interpretation of $\alpha$ is an open question. However, we can consider the physical meaning of the optimized dimensionless number $Z = \theta^\alpha \mathrm{Ca}$, which provides the data collapse for $H$. $Z$ can be decomposed as follows:

$\theta^\alpha \mathrm{Ca} = \left(\frac{\eta/E}{R/V}\right)^\alpha \left(\frac{\rho V^2}{E}\right)^{1-\frac{1}{2}\alpha} = \left(\frac{\mathrm{Relaxation\,time}}{\mathrm{Contact\,time}}\right)^\alpha \left(\frac{\mathrm{Inertial\,force}}{\mathrm{Elastic\,force}}\right)^{1-\frac{1}{2}\alpha}$. Here, we consider $\eta/E$ as the relaxation time associated with the droplet and substrate surface deformation and $R/V$ as the time duration in which a droplet spreads on the substrate, namely the contact time. The time scale ratio can also be expressed using the Deborah number. Therefore, this combination of dimensionless numbers reflects the ratio between the contact time of the droplet and the relaxation time of droplet–substrate deformation, as well as the ratio of substrate elastic force to droplet inertial force. When an increase in droplet viscosity suppresses the spreading of the droplet, the relaxation time becomes significantly longer than the contact time, resulting in the scaling law approaching Hertzian impact scaling. Conversely, when a decrease in droplet viscosity accelerates the droplet spreading, the relaxation time becomes shorter than the contact time, leading to a deviation from Hertzian impact scaling and resulting in inertial force scaling. This is potentially considered a key mechanism for determining the maximum impact force of droplets on elastic substrates. Note that we have defined the relaxation time as the ratio of droplet viscosity to substrate elastic modulus, $\eta/E$, although this quantity is normally expressed using the properties of the same material. The reason for this choice is that the droplet and substrate are always in contact until the maximum impact force appears.

Based on our findings, earlier research has demonstrated that the scaling law for the maximum impact force with a rigid substrate is predominantly governed by Re, which reflects the balance between inertia and viscosity. This approaches $\Pi \propto \mathrm{Ca}$ if the inertia is much greater than the viscosity and is named "inertial force scaling"[15]. In contrast, with an elastic substrate, increasing the viscosity, resulting in an increase in $Z$, shifts the scaling law from inertial force scaling to Hertzian impact scaling. While it is true that high-viscosity droplets naturally yield high $Z$ values (and thus exhibit Hertzian behavior), low-viscosity droplets can also reach high $Z$ if the inertia is sufficiently large. This means that Hertzian impact scaling can emerge even for low-viscosity droplets, provided the impact velocity is high and/or the substrate is sufficiently soft. In other words, Hertzian behavior is not limited to high-viscosity cases, but results from a large value of $Z$, regardless of whether it arises from high $\theta$ or high Ca. This is because the scaling is influenced by the interplay between inertial and elastic forces and the comparison between relaxation time and contact time. Thus, under this condition, i.e., with an elastic substrate, it would be misleading to continue referring to the scaling law $\Pi \propto \mathrm{Ca}$ as "inertial force scaling".

In this paper, we successfully observed the transition of the dynamic behavior of the impacting droplet depending on the droplet viscosity and the substrate elasticity through the measurement of stress and impact force acting on the elastic substrate using the photoelastic tomography technique. We confirmed that this transition is from inertial force scaling to Hertzian impact scaling based on observing the stress distributions and the maximum impact force. Our findings introduce a new insight for predicting the behavior of droplets upon impact with substrates, revealing understandings crucial for both fundamental and applied science. Based on our results—particularly Fig. 3b—we can possibly anticipate whether a droplet will behave like a rigid sphere or remain more fluidic upon impact based on parameters such as viscosity, inertia, and substrate elasticity. This predictive capability has significant implications for applications where impact forces dictate material response. In turbine erosion or water-cutting, low-viscosity water droplets may exhibit behavior like rigid spheres under high inertia, allowing them to erode or cut hard surfaces, including metals. This finding challenges conventional models, emphasizing the need to consider substrate elasticity to accurately predict the droplet impact force. Furthermore, we believe

that these insights have impacted the applications of soft materials. For instance, our ability to assess droplet impact behavior enables more precise control in applications like 3D bioprinting, where impact stress affects the fidelity and viability of printed structures. This study thus marks a step forward in comprehending fluid-structure interactions under high-speed impacts, bridging theoretical advances with practical application across diverse fields Table 1.

## Methods

### Experiment

The experimental setup is shown in Fig. 4a. Droplets of silicone oil (Shin-Etsu, KF-96) with kinetic viscosities of $10^0$, $10^1$, $10^2$, $10^3$, $10^4$ cSt were used. The oil density $\rho$ was successively set to 815.5, 932.2, 962.1, 967.1, and 972.1 kg/m³. The average droplet radius $R$ was 1.27 mm. The rigid sphere was made of plastic and had a radius of $R = 2.98$ mm and a density of $\rho \simeq 1077$ kg/m³. Five gel blocks (Gel I, II, III, IV, and V) made of polyurethane (Exseal Co., Ltd., polyurethane gel phantom, 50 × 50 × 50 mm³) and one gel block (Gel VI) made of gelatin derived from porcine skin (Sigma Aldrich, G6144-1KG) dissolved in pure water at 5 wt% were used as an elastic substrate. Because the low-concentration gelatin gel was fragile, Gel VI was kept in the acrylic

**Table 1 | Elastic moduli and stress-optic coefficients of the substrates used in this study**

| Gel | $E$ (kPa) | $C \times 10^{-9}$ (1/Pa) |
|-----|-----------|---------------------------|
| I | 126.7 | 1.15 |
| II | 108.4 | 1.16 |
| III | 47.4 | 1.07 |
| IV | 44.1 | 1.18 |
| V | 17.9 | 1.10 |
| VI | 2.7 | 38.25 |

container used for its preparation during the experiments. The dimension of gel blocks, e.g., width, is over 10 times larger than the radius of the droplet and the rigid sphere. Therefore, over the short time scales considered in this study, we assume that the substrate can be treated as an elastic half-space, and wall effects, including reflected stress wave, can be ignored. Although the rigid sphere's radius is larger than that of the droplets on average, it remains far smaller than the dimensions of the gel blocks. Consequently, the effect of the impactor radius on the impact force is taken into account in the scaling law by nondimensionalization. Elastic moduli $E$ of each substrate were measured by indentation (see Supplementary Information) and are listed in Table 2.

The droplet or sphere impacted the substrate after falling freely. Impact velocity $V$ was varied from approximately 0.3–3.0 m/s by adjusting the falling height from 1-50 cm. To measure the impact force and stress within the substrate, photoelastic tomography was employed using the high-speed polarization camera (Photron, CRYSTA PI-1P, 20,000 fps) and the light source producing circularly polarized light (Thorlabs, SOLIOS-525C, typical wavelength of $\lambda = 520$ nm). Impact velocity $V$ was measured using the image sequence of the impactor, which is recorded by the high-speed polarization camera, before the impactor touches the substrate. The measurements were repeated three times for each height. The data for each height were averaged and plotted with their standard deviation in the figures. Because Gel VI was fragile, only a single measurement was taken at each height, accordingly, error bars are not shown for those data.

The photoelastic method measures the state of the polarized light passing through the stressed material. This allows the stress field to be evaluated from the optical anisotropy (birefringence) caused by stress loading[46]. When circularly polarized light enters a stressed material, it is modulated as elliptically polarized light with photoelastic parameters (retardation $\Delta$ and orientation $\phi$) relating to the stress state. $\Delta$ and $\phi$ can be measured by a

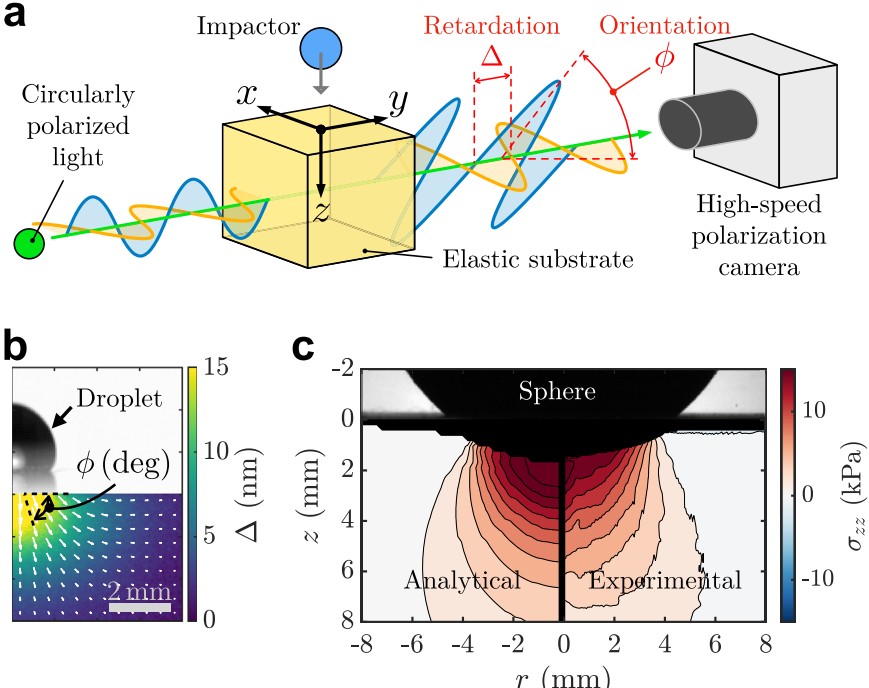

**Fig. 4 | Stress field measurement using high-speed photoelastic tomography. a** Schematic of the experimental setup for measuring the stress field in an elastic substrate during a sphere or droplet impact. **b** Typical image of the photoelastic parameters (retardation $\Delta$ and orientation $\phi$) during the droplet impact. The color indicates $\Delta$ and the white arrows indicate $\phi$. The length of white arrows corresponds to the magnitude of retardation. **c** Comparison of the axial stress within the substrate, Gel III (see Table 2), induced by the contact of the styrol sphere obtained from Hertzian contact theory[34,67] (left) and from photoelastic tomography (right). The diameter of sphere and contact force are 14.7 mm and 0.55 N, respectively.

**Table 2 | Elastic moduli and stress-optic coefficients of the substrates used in this study**

| Gel | $E$ (kPa) | $C \times 10^{-9}$ (1/Pa) |
|---|---|---|
| I | 126.7 | 1.15 |
| II | 108.4 | 1.16 |
| III | 47.4 | 1.07 |
| IV | 44.1 | 1.18 |
| V | 17.9 | 1.10 |
| VI | 2.7 | 38.25 |

polarization camera based on the four-step phase-shifting method[51,66]. Four linear micro-polarizers were installed in neighboring pixels of the camera's image sensor. The angles of the linear polarizers were set to 0°, 45°, 90°, and 135°, and the corresponding camera sensor measures the light intensity through these linear polarizers, denoted by $I_{0°}, I_{45°}, I_{90°}$, and $I_{135°}$, respectively. $\Delta$ and $\phi$ can be obtained from the four intensity values as follows: $\Delta = \frac{\lambda}{2\pi}\sin^{-1}\frac{\sqrt{(I_{90°}-I_{0°})^2+(I_{45°}-I_{135°})^2}}{(I_{0°}+I_{45°}+I_{90°}+I_{135°})/2}$, $\phi = \frac{1}{2}\tan^{-1}\frac{I_{90°}-I_{0°}}{I_{45°}-I_{135°}}$. Spatial resolutions of the measured photoelastic parameters' image are 48.0 and 83.4 μm/pix for droplet and sphere cases, respectively. A typical image of $\Delta$ and $\phi$ during droplet impact is shown in Fig. 4b. The relationship between the stress field and photoelastic parameters is called the stress-optic law[46]. $\Delta$ is proportional to the integration of the secondary principal stress difference along the camera's optical axis, and $\phi$ is related to the direction of the secondary principal stress[46]. This relationship can be expressed as follows: $\Delta\cos 2\phi = C\int_{-\infty}^{\infty}(\sigma_{xx}-\sigma_{zz})dy$, $\Delta\sin 2\phi = 2C\int_{-\infty}^{\infty}\sigma_{xz}dy$, where $C$ is the stress-optic coefficient. The stress-optic coefficient $C$ for each substrate is listed in Table 2. The stress components in Cartesian coordinates are $\sigma_{xx}$, $\sigma_{zz}$, and $\sigma_{xz}$, with the $y$-axis as the camera's optical axis, as shown in Fig. 4. From $\Delta$ and $\phi$, the dynamic stress field in the substrate, $\sigma_{zz}(x, z, t)$, can be reconstructed using our recently developed high-speed photoelastic tomography technique[36]. To validate our photoelastic tomography, we measured the axial stress field induced by the static contact of a rigid sphere and compared it with Hertzian contact theory[34,67]. In Fig. 4c, we demonstrate an example of the validation using the substrate, Gel III. Experimental measurements agree well with the numerical results, validating the accuracy of photoelastic tomography. More details of the reconstruction algorithm are described in the Supplementary Information.

As can be seen from the image on the right in Fig. 4b, the light intensity below the horizontal position, where the droplet contacts the substrate surface, varies due to the reflection of droplets on the substrate surface. It leads that the photoelastic parameters measured in this area are inaccurate. To exclude this region, in the stress field reconstruction, we defined the position 0.7 mm below the contact position between the droplet and the substrate as $z = 0$ and performed the analysis.

The impact force $F$ can be estimated by integrating the reconstructed $\sigma_{zz}$ at $z = 0$, i.e., $F(t) = 2\pi\int_0^{\infty}\sigma_{zz}rdr$. Because reconstructed $\sigma_{zz}$ decreases with $r$, retardation $\Delta$ becomes small and orientation $\phi$ randomizes at large $r$, increasing noise in $\sigma_{zz}$. Since the integrand $\sigma_{zz}r$ further amplifies noise at large $r$, we limit the upper integral bound to the dynamic contact radius $r_c \simeq 2R\sqrt{tV/R}$[68]. Consequently, $F(t) = 2\pi\int_0^{r_c}\sigma_{zz}(r, t)rdr$ was used in the data processing.

### Prayer beads algorithm for data collapse

To identify the self-similar solution to describe the crossover of scaling law[65,69], here we briefly review an algorithm to get data collapse.

Suppose that there are physical parameters $u, q, T$ and a function $u = f(q, T)$. Here, we think about the operation to change the scale of

parameters preserving the similarity; it is called scale-transformation, e.g., $(u\prime, q\prime, T\prime) = (Au, A^\kappa q, A^\lambda T)$ where $A$ is a factor and $\kappa, \lambda$ are power exponents. We can deduce invariant functions of the scale-transformation, which is the form of the function such that does not vary by the scale-transformation. It was known that invariant functions of scale-transformation are always power-law monomials[64,70], e.g., $q/u^\kappa$, $T/u^\lambda$. Therefore, when the parameters of the function are replaced by the invariant form such as $q/u^\kappa = \Phi(T/u^\lambda)$ where $\Phi$ is called a scaling function, and $q/u^\kappa$ and $T/u^\lambda$ are called similarity parameter, the function $\Phi$ does not change by the scale-transformation as well. It means that replotting the data points by using similarity parameters renormalizes the difference of the scale of the parameters to all the data points converge to a single line described by $\Phi$; it is data collapse, which is the expression exploiting the self-similarity and offering the fundamental insight about the problem.

This insight suggests that the condition of data collapse can be realized by identifying the power exponents of similarity parameters. Dimensional analysis is a powerful method to get power exponents, which is called similarity of the first kind, though it is only applicable to special cases. Because the problems belong to self-similarity parameters of the second kind, in which the dimensional analysis cannot deduce the power exponents. In such a case, the data collapse relation can be obtained by optimizing an appropriate evaluation function of the power exponents, $I(\kappa, \lambda)$

Here, we propose a method to determine $\kappa$ and $\lambda$ using the data-driven "prayer beads algorithm". When $\kappa$ and $\lambda$ are selected appropriately, all the scattering data points converge to a single curve described by the self-similar function $\Phi$. At this time, the total distance between neighboring data points is minimized. Based on this observation, we define the evaluation function as the distance to each data point. Here, either $\kappa$ or $\lambda$ must be fixed to prevent data points from concentrating on a single point. This picture corresponds to pulling the rope of relaxed prayer beads. Supposing an exponent of one similarity parameter $\kappa$ is known a priori to be $\kappa^*$, the evaluation function $I$ that optimizes $\lambda$ using $N$ data points can be written as follows:

$$I(\lambda) = \sum_i^N \sqrt{(\log(q_{i+1}/u_{i+1}^{\kappa^*}) - \log(q_i/u_i^{\kappa^*}))^2 + (\log(T_{i+1}/u_{i+1}^\lambda) - \log(T_i/u_i^\lambda))^2}.$$

(5)

The optimized $\lambda^*$ is obtained when $I(\lambda^*) = \min I(\lambda)$. In this study, the evaluation function of the problem was $I(\alpha) = \sum_i^N \sqrt{(\log(H_{i+1}/H_i))^2 + (\log(Z_{i+1}/Z_i))^2}$, where $H = \Pi/Ca^{3/5}$ and $Z = \theta^\alpha Ca$.

## Data availability

The data supporting the findings of this study are provided in the Source Data file. Any additional data, including raw high-speed polarization imaging datasets, are available from the corresponding authors upon request due to their large file size. Source data are provided with this paper.

## Code availability

The code used for the prayer beads algorithm is provided in the Supplementary Materials.

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

## Acknowledgements
We thank Dr. D. Lohse, Dr. V. Sanjay, and Dr. S. Mandeep for their valuable discussions and suggestions. We thank W. Worby for the AFM measurement to estimate the substrate roughness. This work was supported by JSPS KAKENHI Grant Numbers JP24H00289 (Y.T.), JP24KJ2176 (Y.Y.), JP22KJ1239 (Y.Y.), and JST PRESTO Grant Number JPMJPR21O5 (Y.T.), Japan.

## Author contributions
Y.Y. and Y.T. convinced and designed the project. Y.Y. and K.M. performed the experiments. Y.Y., K.M., and H.M. analyzed the data. Y.Y., K.M., H.M., and Y.T. wrote the manuscript.

## Competing interests
The authors declare no competing interests.
