## [Transparent Peer Review file · Nature Communications]

Scaling Crossover in Droplet Impact Force on Elastic Substrates

Corresponding Author: Professor Yoshiyuki Tagawa

Version 0:

Reviewer comments:

Reviewer #1

(Remarks to the Author)

In this manuscript, the authors investigate and reveal a unique pattern transition of the drop impact dynamics on an elastic substrate via an advanced photo-elastic tomography technique. The paper's overall scope is well-justified and should appeal to a broad readership, particularly in the fields of fluid dynamics and interfacial science. The central discovery of this paper highlights that the impacting pattern shows a transition from inertial force scaling to Hertzian impact scaling as a function of drop viscosity and substrate stiffness via the observation of stress distributions and maximum impact forces. This finding extends previous research by accounting for substrate elasticity, which was overlooked in favor of inertial force scaling. The scaling model in this work is well presented and nicely compared to the experimental data. Nonetheless, a number of key issues listed below, both experimental and theoretical, should be resolved and clarify.

1. In line 62, the authors claimed that the forces and stresses dynamics generated during droplet collisions remain poorly understood. However, in recent years, the contents have been reported in [Physical Review Letters, 2022, 129(10), 104501] and [Annual Review of Fluid Mechanics, 2022, 54(1): 57-81], et al.
2. In line 83, the authors suddenly claimed that "Additionally, if the substrate is elastic, it can be supposed that the substrate deformation also becomes significant as viscosity increases. Therefore..." The authors should make a less confusing and more logical hypothetical statement.
3. In the present manuscript, the authors conducted a series of studies employing rigid sphere and viscous drops to compare the impact behaviors. However, the radius of the rigid sphere (2.98 mm) differs from that of the viscous drops (1.27 mm). Although the results are presented in dimensionless form, the authors should provide a rationale for using different impactor radii and explicitly clarify this discrepancy.
4. The paper considers a broad range of impact heights. The authors should specify whether impact velocities were calculated from free-fall heights or directly measured from the impacting videos. If the former approach was used, they should also evaluate the potential influence of velocity measurement errors on their results.
5. In this manuscript, the authors investigate the effects of drop viscosity and substrate stiffness on impact dynamics, focusing primarily on drops impacting a soft substrate with an elastic modulus of 47.4 kPa. However, substrate deformation is also critical in influencing impact transitions. Have the authors explored varying the substrate elasticity to further understand its role?
6. In Figure 1, the stress orientation (white arrows) at $t = 1.0$ ms for the drop of 10000 cSt appears unusual, as it is significantly different from the expected pattern during the spreading phase. Could the authors clarify this discrepancy in the manuscript?
7. In Figure 2, it is recommended to move the diagram for the 100 cSt case to the supplementary materials.
8. In line 241, the authors note that $F(t)$ is symmetrical for sufficient high-impact velocity and viscosity drops. Conversely, for low viscosity drops, $F(t)$ is asymmetric. However, this symmetry was observed only for $t < 1$ ms (Figure 2b). It is recommended to specify the time period in the text.
9. The authors engage in a thorough discussion of two scaling laws for impact forces: inertial impact and Hertzian impact. However, the authors' analysis would be strengthened by the inclusion of critical conditions for the transition between these two scaling laws. This is particularly important for the selection of drops and substrates in relevant applications.
10. Line 340 states, "In contrast, with an elastic substrate, increasing the inertia resulting in an increase in Z , shifts the scaling law from inertial force scaling to Hertzian impact scaling, even if the droplet viscosity is low." This assertion is perplexing. According to Figure 3, the Hertzian impact only occurs for high viscosity drops.

Reviewer #2

(Remarks to the Author)

Droplet impact on elastic substrates: crossover scaling in impact forces

This study investigates the dynamics of droplet impact on elastic substrates, with a focus on how impact forces scale under different conditions. The authors visualize the dynamic stress fields in soft materials by using high-speed photoelastic tomography and uncover a crossover in impact force scaling. Their findings reveal that as substrate elasticity increases, the maximum impact force shifts from inertial force scaling to Hertzian impact scaling. This transition is governed by a newly identified similarity parameter, which accounts for droplet inertia, viscosity, and substrate deformation timescales. The authors propose a data-driven framework for predicting impact forces based on viscosity, inertia, and elasticity, offering practical guidance for engineering and scientific applications.

I found the manuscript interesting and worthy of the novelty, however there are some major issues that should be addressed.

List of comments:

1. The authors should improve the clarity and consistency of data presentation while ensuring uniform formatting throughout the manuscript. For instance, fig. 1 and fig. 3 use different fonts. Additionally, fig. 1 lacks a scale bar, making it difficult to gauge spatial dimensions (for example the deformation of the soft substrate). The color bar in fig. 1 is not intuitive, which affects readability and make the interpretation of results challenging. The authors should consider using diverging color schemes to enhance visual clarity and make the data more accessible to readers.

2. The authors indicates that the white arrows in fig.1 represent \emptyset , which is related to the direction of the secondary principal stress. However, the arrows appear to vary in size. Could the authors clarify whether the size of the white arrows corresponds to the magnitude of the secondary principal stress or if it serves a different purpose?

3. On page 5 at Line 223-224, the authors claim that negative stress is more pronounced in the sphere case at 1.0 ms in fig. 1. But it is not clear from the figure. Could authors shed light on this and explain this phenomenon with more clarity?

4. A drop impacting a soft surface can lead to the formation of an out-of-plane wetting ridge at the three-phase-contact line. Could the authors elaborate on the phase separation dynamics within this wetting ridge and its key characteristics? How does the extent of phase separation vary with silicone oil viscosity, and what influence does the emergence of the wetting ridge have on the overall stress distribution in the substrate?

5. Since the authors have used soft gel with elastic modulus of 47.4 kPa, Could the authors provide a discussion on the dynamics of stress generation due to the deformation of the three-phase contact line (wetting ridge)? How does this deformation influence the overall impact force and stress distribution in the substrate?

6. On page 6, Line 234-238, the authors vaguely speculate and does not provide any evidence or analysis to support their claim regarding the observation of central stress peak. Furthermore, the authors appear to misinterpret the findings of Sun et al., who demonstrated that for a droplet impacting an elastic substrate, the stress peak (σ_{zz}) remains non-central, rather than shifting toward the center ($r = 0$). The authors could substantiate their claim by demonstrating that the observations remain consistent across substrates with different elastic moduli? This would help confirm that the result is not merely an experimental artifact.

7. It has been reported in literature that a droplet impacting a soft surface can entrap more air compared to a rigid flat surface, with the size of the entrapped air decreasing based on impact velocity and droplet viscosity. Could the authors comment on whether variations in the air layer thickness beneath the droplet, influenced by different viscosities and substrate softness, affect the stress field distribution? It would be a much stronger story if the elastic modulus of the substrate were varied as well.

8. The authors can include the relevant literature and cite appropriately, specifically, the paper "Understanding the drop impact phenomenon on soft PDMS Substrate" by S. Mangily and "Droplet impacts onto soft solids entrap more air" by Langely that would strengthen the context of their study and provide a more comprehensive background and situate the current work within the broader research landscape.

Reviewer #3

(Remarks to the Author)

The authors of the manuscript conducted an investigation on the impact forces of liquid droplets on elastic solids, wherein a transition from the inertial regime to the Hertzian regime has been observed. Specifically, they employed the high speed photoelastic tomography to measure the stress on an elastic substrate which was imparted by impacting droplets of different velocities and viscosity. Though the manuscript is well written and organized, it does not meet the requirements of the journal due to the following issues.

1. The impact forces of liquid droplets on solid surfaces have been extensively investigated over the past few years (Soft matter 10 27, 4929-4934 2014; Physical Review Letters 12910, 104501 2022; Phys. Fluids 35, 052104 2023; Physics of Fluids 35, 112111 2023; Phys. Rev. Fluids 8, 113601 2023). However, relevant key findings were not adequately discussed

and cited. Moreover, the topic of this study is too specific to attract broad readerships of Nature Communications.

2. The high speed photoelastic tomography needs to be validated by with other sophisticated force measurement method or numerical simulations. As simple comparison between the results of this work and that of Soto et al. (Soft matter 10 27, 4929-4934 2014), the measured maximum impact forces by the authors are much smaller.
3. Only one type of elastic substrates ($E=47.4$ kPa) has been used in this study, which is thus insufficient to draw the general conclusion as the authored did.
4. Previous studies have demonstrated that hydrodynamic forces applied on solid surfaces are highly dependent on the drop behaviors (Phys. Fluids 35, 052104 2023; Physics of Fluids 35, 112111 2023; Phys. Rev. Fluids 8, 113601 2023). However, the effects of droplet dynamics on the impact forces were not analyzed at all.
5. Early studies have explicitly demonstrated that there is a liquid hammer pressure (e.g., Phys. Med. Biol. 36, 1475 1991; J. Fluid Mech. 490, 1 2003) up impacting on solid surfaces. This impact force is much larger than the hydrodynamic force investigated here, and it is much more important in practical applications, e.g., causing the well-known raindrop erosion. Therefore, the authors should include this force term in their studies.

Version 1:

Reviewer comments:

Reviewer #1

(Remarks to the Author)

I appreciate all the changes made by the authors and I am pleased to see the publication of this paper.

Reviewer #2

(Remarks to the Author)

I am satisfied with the authors responses to my questions and appreciate their care and extra experiments that were required to address them. I recommend publication.

Reviewer #3

(Remarks to the Author)

From the rebuttal one can read that the authors have managed to improve the manuscript following the suggestions of three reviewers. However, some key issues need to be further addressed before considering for publication.

1. All three reviewers have suggested to perform additional measurements on other soft substrates to demonstrate the robustness of the finding. Though new experimental results were added in the revised manuscript, they were obtained from solid substrates with comparable moduli ($E=44$ kPa and 17.9 kPa) as that in the previous version ($E=47.4$ kPa). Instead, the authors should conduct experiments on solid substrates with modulus difference of orders of magnitude. From Fig. 3a, it can be found that the experimental data already exhibits strong scattering, which suggests the importance of conducting experiments on a wide range of solid substrates.

2. Compared to existing results of measuring impact forces of liquid droplets, the authors have ascribed the absence of a second peak value to that no bouncing phenomenon occurred in their experiments. However, the fact is that the second peak force in droplet impact is produced by the cavity collapse (Physical Review Letters 12910, 104501 2022; Phys. Fluids 35, 052104 2023; Physics o Fluids 35, 112111 2023; Phys. Rev. Fluids 8, 113601 2023), which almost happens in all impact events regardless of the viscosity of the liquid and wetting property of the solid substrate (Annual Review of Fluid Mechanics 48, 365–391 2016; Physical Review Letters 96, 124501 2006). This inconsistency should be carefully explained as it relates to reliability of the force/stress measurement methodology.

3. The effect of air cushion on the stress distribution has been discussed; however, no explicit comments/conclusions were given/made. The appearance of an air cushion below impacting liquid droplets is a ubiquitous phenomenon, and it can even happen on very rigid solid substrates, making droplets bouncing off (Nature Physics 11, 48–53 2015). Therefore, it is very crucial to clarify whether it affects the impact stress applied on the solid substrates. Actually, performing a comparative experiments of droplet impact in air and in vacuum chamber can provide some clues. Regarding the second issue, does the absence of the second peak force result from the air cushion?

Version 2:

Reviewer comments:

Reviewer #3

(Remarks to the Author)

The authors have carefully addressed the remaining issues and the work is suitable for publication.

We would like to thank the referee for the constructive comments and attention to detail, which from our point of view, have led to further improvement of the manuscript. The text color of the changes are shown in red in the marked-up manuscript.

Reviewer #1

Summary:

In this manuscript, the authors investigate and reveal a unique pattern transition of the drop impact dynamics on an elastic substrate via an advanced photo-elastic tomography technique. The paper's overall scope is well-justified and should appeal to a broad readership, particularly in the fields of fluid dynamics and interfacial science. The central discovery of this paper highlights that the impacting pattern shows a transition from inertial force scaling to Hertzian impact scaling as a function of drop viscosity and substrate stiffness via the observation of stress distributions and maximum impact forces. This finding extends previous research by accounting for substrate elasticity, which was overlooked in favor of inertial force scaling. The scaling model in this work is well presented and nicely compared to the experimental data. Nonetheless, a number of key issues listed below, both experimental and theoretical, should be resolved and clarify.

Thank you for your valuable feedback on our manuscript. We have carefully considered your comments and have made revisions to address each of your concerns, as outlined below.

1. In line 62, the authors claimed that the forces and stresses dynamics generated during droplet impacts remain poorly understood. However, in recent years, the contents have been reported in [Physical Review Letters, 2022, 129(10), 104501] and [Annual Review of Fluid Mechanics, 2022, 54(1): 57-81], et al.

Thank you for your helpful comment. We agree that the original phrase, “remain poorly understood,” may have unintentionally understated the progress made in recent years on droplet impact force and stress dynamics. In response, we have revised the sentence to more appropriately reflect the current state of the field.

In accordance with your comment, we have revised the relevant sentence in the Introduction to replace “poorly understood” with “less explored experimentally for soft substrates” and moved the sentence to line 116 after reviewing the previous studies about droplet impact on flexible substrates. Additionally, we have clarified the position of our work relative to prior studies with appropriate citations.

2. In line 83, the authors suddenly claimed that “Additionally, if the substrate is elastic, it can be supposed that the substrate deformation also becomes significant

as viscosity increases. Therefore. . . .” The authors should make a less confusing and more logical hypothetical statement.

Thank you for pointing this out. In the original manuscript, the hypothesis that higher viscosity leads to greater substrate deformation was unclear. To make the expression more logical, we have revised the relevant section as follows:

“These findings imply that a soft substrate can alter droplet deformability and thus influence F_{\max} . When the substrate is sufficiently soft, substrate deformation may suppress droplet deformation. This effect is expected to be more pronounced as the droplet viscosity increases.”

3. In the present manuscript, the authors conducted a series of studies employing a rigid sphere and viscous drops to compare the impact behaviors. However, the radius of the rigid sphere (2.98 mm) differs from that of the viscous drops (1.27 mm). Although the results are presented in dimensionless form, the authors should provide a rationale for using different impactor radii and explicitly clarify this discrepancy.

Thank you for pointing this out. Due to experimental constraints, it was technically challenging to conduct impact experiments using a rigid sphere with a radius as small as that of the droplet (approximately 1.27 mm). Therefore, we used a larger rigid sphere with a radius of 2.98 mm. Nevertheless, the rigid sphere remains more than 10 times smaller than the gel substrate ($50 \times 50 \times 50, \text{mm}^3$), satisfying the condition for treating the substrate as a semi-infinite elastic body. This ensures the applicability of Hertzian impact theory and allows us to neglect boundary effects, such as reflected stress waves.

As the reviewer correctly pointed out, we present and analyze our results in terms of dimensionless quantities. These quantities, derived via dimensional analysis, explicitly incorporate the impactor radius and normalize differences in length scale. Therefore, we believe that comparisons of scaling relationships expressed in this non-dimensional form remain valid even when using impactors of different radii.

Moreover, since the 2.98 mm rigid sphere already satisfies the semi-infinite substrate assumption, a smaller 1.27 mm rigid sphere—if it had been used—would have made the relative size of the substrate even larger. Thus, the condition for semi-infinite elasticity would be better satisfied. The radius difference in our experiment does not affect the qualitative nature of the impact mechanics in the Hertzian regime. In light of this, we have revised the Method section to explicitly state the difference in radii and to clarify the rationale behind our comparison using dimensionless quantities.

4. The paper considers a broad range of impact heights. The authors should specify whether impact velocities were calculated from free-fall heights or directly measured from the impacting videos. If the former approach was used, they should also evaluate the potential influence of velocity measurement errors on their results.

Thank you for pointing this out. Regarding the measurement method for impact velocity, as you asked, we did not calculate it from the free fall height but instead directly measured the velocity just before the impactor contacted the substrate from the image recorded by the high-speed polarized camera. This minimizes velocity errors caused by air resistance during the fall. This information has been added to the Method section of the manuscript.

5. In this manuscript, the authors investigate the effects of drop viscosity and substrate stiffness on impact dynamics, focusing primarily on drops impacting a soft substrate with an elastic modulus of 47.4 kPa. However, substrate deformation is also critical in influencing impact transitions. Have the authors explored varying the substrate elasticity to further understand its role?

Thank you for pointing this out. The experimental data included in the original manuscript used only a single substrate elasticity (denoted as Gel I in the revised manuscript). As you suggested, we believe that experiments with different substrate elasticities are essential to strengthen the conclusion, so we conducted additional experiments using substrates with different elasticities (denoted as Gel II and III). Additionally, we reanalyzed the experimental data of Gel I for consistency. Table 1 in the manuscript summarizes the elasticity E of each gel together with the stress-optic coefficient C used for the stress reconstruction. Additionally, we have added the methods for determining E and C for each substrate to the Supplementary Information. The results of the additional experiments confirmed that, in the scaling laws of Π and Ca , the transition from inertial force scaling to Hertzian impact scaling occurs with increasing droplet viscosity, even for substrates with different elastic moduli. To quantify the changes in scaling laws more precisely, we added the scaling law's prefactor and index for Π vs Ca for each substrate elasticity-droplet viscosity combination to the Supplementary Information (Fig. 6). In the graph of H vs Z (Fig. 3 of the manuscript), the crossover phenomenon and data collapse due to the similarity parameter Z were still observed. Notably, in experiments using the less elastic Gel III, droplets of silicone oil with a viscosity of 10^4 cSt exhibited behavior close to Z^0 , resembling the behavior observed in the sphere case. This further strengthens the universality of the scaling relationship proposed in this study. The results of this additional experiment have been reflected in the figure, and the cor-

responding descriptions have been added to section 2 in the revised manuscript.

6. In Figure 1, the stress orientation (white arrows) at $t = 1.0$ ms for the drop of 10000 cSt appears unusual, as it is significantly different from the expected pattern during the spreading phase. Could the authors clarify this discrepancy in the manuscript?

Thank you for your comment. The behavior of the stress direction (white arrows) at $t = 1.0$ ms in the 10^4 cSt droplet, which appears to differ from that in the droplet spreading stage, is explained below. First, at $t \sim 1.0$ ms, the spreading of the highly viscous 10^4 cSt droplet is almost complete. At this point, the substrate, which has been slightly deformed downward due to the impact, begins to return to its original position due to its own elasticity. This results in the generation of negative stress in the lower region of the droplet-substrate contact area (see also Fig. 2(b) at $t = 1$ ms for the 10^4 cSt droplet). This negative stress is also observed at later times after a rigid sphere impacts with an elastic substrate (see Fig. 2(b) in the revised manuscript at $t = 2$ ms for the sphere, and Supplementary Information Fig. 2). This represents one example of the behavior of high-viscosity droplets, which resembles that of rigid spheres. In the 10^4 cSt silicone oil droplets, the arrows near the central axis differ from those in other droplets, and this is primarily due to the relatively large negative stress characteristic of the 10^4 cSt silicone oil droplets. However, in the original figure, the lengths of the arrows were rescaled according to the magnitude of the retardation distribution at each time step, which was not intuitive and led to misunderstanding. In the revised Fig. 1, which was updated based on the comments of other reviewers, the plot has been revised using a consistent length scale for the arrows. In addition, explanations regarding the negative stress of 10^4 cSt and the negative stress at a later time of the sphere were added to the manuscript.

7. In Figure 2, it is recommended to move the diagram for the 100 cSt case to the supplementary materials.

Thank you for your suggestion. We have revised Fig. 2 to show only the two extreme experimental data sets (10^4 cSt and 10^0 cSt). The data for the 100 cSt case has been moved to the Supplementary Information (Fig. 3).

8. In line 241, the authors note that $F(t)$ is symmetrical for sufficient high-impact velocity and viscosity drops. Conversely, for low viscosity drops, $F(t)$ is asymmetric. However, this symmetry was observed only for $t \geq 1$ ms (Figure 2b). It is recommended to specify the time period in the text.

Thank you for your suggestion. Regarding the description of the symmetry of $F(t)$, as shown in Fig. 2(b), this symmetry is indeed observed mainly in the short time scale near the peak immediately after the impact. To provide a more accurate representation of the situation, we have added a description indicating the time range observed at the relevant location in section 2.1.

9. The authors engage in a thorough discussion of two scaling laws for impact forces: inertial impact and Hertzian impact. However, the authors' analysis would be strengthened by the inclusion of critical conditions for the transition between these two scaling laws. This is particularly important for the selection of drops and substrates in relevant applications.

Thank you for your suggestion. Identifying the critical conditions at the transition point of the force scaling law is important not only for fundamental research but also for practical applications. Since the crossover of scaling laws is a phenomenon in which two asymptotic behaviors change continuously, it is difficult to specify a clear critical value in the strict sense. However, by using the similarity variable $Z = \theta^\alpha Ca$, it is possible to propose a practical threshold of behavior based on a characteristic Z value (Z_c), where $Z \gg Z_c$ corresponds to behavior closer to the Hertz impact scaling law and $Z \ll Z_c$ corresponds to behavior closer to the inertial force scaling law. Based on the results of this study, Z_c is approximately 1. Using this value of Z can help predict which scaling law will follow based on parameters such as droplet viscosity, velocity, and substrate elasticity. We have added explanations to section 2.2 of the manuscript and indicated the proposed value of Z_c .

10. Line 340 states, "In contrast, with an elastic substrate, increasing the inertia resulting in an increase in Z , shifts the scaling law from inertial force scaling to Hertzian impact scaling, even if the droplet viscosity is low." This assertion is perplexing. According to Figure 3, the Hertzian impact only occurs for high viscosity drops.

Thank you for your comment. We understand the confusion and appreciate the opportunity to clarify. The key point is that the transition to Hertzian impact scaling is governed not by viscosity alone, but by the similarity parameter $Z = \theta^\alpha Ca$, which depends on both the viscous effect (θ) and inertia (Ca).

While it is true that high-viscosity droplets naturally yield high Z values (and thus exhibit Hertzian behavior), low-viscosity droplets can also reach high Z if the inertia is sufficiently large. This means that Hertzian impact scaling can emerge even for low-viscosity droplets, provided the impact velocity is high and/or the substrate is sufficiently

soft. In other words, *Hertzian behavior is not limited to high-viscosity cases*, but results from a large value of Z , regardless of whether it arises from high θ or high Ca .

To avoid misunderstanding, we have revised Section 2.2 to clarify that it is the magnitude of Z , not viscosity alone, that governs the observed crossover in scaling behavior.

We would like to thank the referee for the constructive comments and attention to detail, which from our point of view, have led to further improvement of the manuscript. The text color of the changes are shown in red in the marked-up manuscript.

Reviewer #2

Summary:

This study investigates the dynamics of droplet impact on elastic substrates, with a focus on how impact forces scale under different conditions. The authors visualize the dynamic stress fields in soft materials by using high-speed photoelastic tomography and uncover a crossover in impact force scaling. Their findings reveal that as substrate elasticity increases, the maximum impact force shifts from inertial force scaling to Hertzian impact scaling. This transition is governed by a newly identified similarity parameter, which accounts for droplet inertia, viscosity, and substrate deformation timescales. The authors propose a data-driven framework for predicting impact forces based on viscosity, inertia, and elasticity, offering practical guidance for engineering and scientific applications.

I found the manuscript interesting and worthy of the novelty, however there are some major issues that should be addressed.

Thank you for your valuable feedback on our manuscript. We have carefully considered your comments and have made revisions to address each of your concerns, as outlined below.

1. The authors should improve the clarity and consistency of data presentation while ensuring uniform formatting throughout the manuscript. For instance, fig. 1 and fig. 3 use different fonts. Additionally, fig. 1 lacks a scale bar, making it difficult to gauge spatial dimensions (for example the deformation of the soft substrate). The color bar in fig. 1 is not intuitive, which affects readability and make the interpretation of results challenging. The authors should consider using diverging color schemes to enhance visual clarity and make the data more accessible to readers.

Thank you for your suggestion. We have standardized the font style across all figures and added a scale bar to Fig. 1. Furthermore, we revised the colormap in Fig. 1 by re-evaluating the color map to improve visibility, as shown in the figure below, and facilitate intuitive interpretation of the stress distribution.

2. The authors indicates that the white arrows in fig.1 represent ϕ , which is related to the direction of the secondary principal stress. However, the arrows appear to vary in size. Could the authors clarify whether the size of the white arrows

Figure 1: **Dynamic behaviors of the impactors and the substrate response.** Spatio-temporal distribution of the stress field (bottom-left panels) photoelastic parameters, Δ and ϕ , and (bottom-right panels) reconstructed axial stress σ_{zz} when a sphere and droplet impact on the elastic substrate, Gel I ($E = 47.4$ kPa) with $V \simeq 2.8 \pm 0.1$ m/s. In the bottom-left panels, the colormap indicates the retardation Δ , and white arrows indicate the orientation ϕ . The arrow length corresponds to the magnitude of retardation shown in the colorbars. The gray-scale images of the impactor on the upper panel are obtained by subtracting the background image without the impactor.

corresponds to the magnitude of the secondary principal stress or if it serves a different purpose?

Thank you for your comment. In Fig. 1, the white arrows indicate the orientation angle measured by the polarization camera. The length of each arrow corresponds to the retardation magnitude, normalized for each droplet viscosity case. In the revised figure, to support a more intuitive understanding, we used a consistent length scale for arrows across different viscosities (e.g., at 0.2 ms, the arrow for 10^4 cSt is longer than that for 10^0 cSt). We have added an explanation of these points to the Fig. 1 caption and the relevant section in the revised manuscript.

3. On page 5 at Line 223-224, the authors claim that negative stress is more pronounced in the sphere case at 1.0 ms in fig. 1. But it is not clear from the figure. Could authors shed light on this and explain this phenomenon with more clarity?

Thank you for pointing this out. The image of the sphere case in Fig. 1 only focuses on the initial period immediately after the impact, so the negative stress was not captured. This negative stress occurs during the process of the elastic substrate recovering from deformation caused by the impact. In cases like sphere impacts that involve significant deformation, it becomes more pronounced at a later time compared to droplet impacts. To clarify this phenomenon, we have added a figure to the Supplementary Information, as shown in the figure below, that shows the reconstructed stress field at a later time after the sphere impact. This allows the more pronounced negative stress in the case of sphere impacts to be visually confirmed. The description in section 2.1 has also been revised to refer to this supplementary figure (Fig. 2), making it clearer.

4. A drop impacting a soft surface can lead to the formation of an out-of-plane wetting ridge at the three-phase-contact line. Could the authors elaborate on the phase separation dynamics within this wetting ridge and its key characteristics? How does the extent of phase separation vary with silicone oil viscosity, and what influence does the emergence of the wetting ridge have on the overall stress distribution in the substrate?

Thank you for your question regarding the wetting ridge and the dynamics of phase separation. When a droplet lands on a soft elastic substrate, the substrate near the contact line is vertically deformed by the surface tension of the droplet, forming a wetting ridge [1]. The characteristic scale of this deformation is given by the elasto-capillary length $l_e = \gamma/E$ [2, 3, 4]. For instance, using the softest substrate in our study ($E = 17.9$ kPa) and the surface tension of silicone oil ($\gamma \approx 20$ mN/m), the wetting ridge height is estimated to be approximately $1.2 \mu\text{m}$, which is below the resolution of our optical system. For a stiffer

Figure 2: **Dynamic behaviors of the rigid sphere and the substrate response.** Spatio-temporal distribution of the stress field (bottom-left panels) photoelastic parameters, Δ and ϕ , and (bottom-right panels) reconstructed axial stress σ_{zz} when a rigid sphere impacts on the elastic substrate, Gel I ($E = 47.4$ kPa) with $V \simeq 2.8 \pm 0.1$ m/s. In the bottom-left panels, the colormap indicates the retardation Δ and white arrows indicate the orientation ϕ . The length of white arrows regard to the magnitude of retardation shown in the colorbar and corresponds to the one in Fig 1 of the main text. The gray-scale images of the impactor on the upper panel are obtained by subtracting the background image without the impactor.

substrate ($E = 47.4$ kPa), the height is estimated to be around $0.5 \mu\text{m}$. In general, on dry soft solids, the ridge consists of a uniform phase. However, if the substrate is swollen with a liquid, capillary forces can induce *phase separation* within the ridge [5]. This separation creates a pure liquid phase localized near the contact line and depends on both the swelling ratio and viscosity of the infiltrating liquid. The growth of this liquid phase occurs over a long timescale (tens to thousands of seconds), and its dynamics are significantly slowed in the case of higher viscosity oils or low swelling ratios. Under dynamic conditions like droplet impact, this phase separation is known to be suppressed due to insufficient time for separation to occur [4]. Thus, we expect this effect to be negligible during the timescale of our impact experiments. These points have been discussed in the Supplementary Information.

5. Since the authors have used soft gel with elastic modulus of 47.4 kPa. Could the authors provide a discussion on the dynamics of stress generation due to the deformation of the three-phase contact line (wetting ridge)? How does this deformation influence the overall impact force and stress distribution in the substrate?

Thank you for this important question. The vertical force exerted on the substrate by the wetting ridge can be estimated from the surface tension and the contact line length. For a typical silicone oil droplet with radius 1 mm and surface tension $\gamma \approx 20$ mN/m, this force is estimated to be around 0.13 mN (assuming a contact angle of 90°). This is more than 10 times smaller than the lowest maximum impact force ($F_{\text{max}} = 2$ mN) measured in our experiments.

Moreover, the influence of the wetting ridge on the stress distribution is limited to a region of approximately one elasto-capillary length ($\sim 1 \mu\text{m}$) around the contact line. Therefore, we consider its contribution to the overall stress field during impact to be very localized and minor. That said, in dynamic wetting processes, the deformation induced by the wetting ridge has been suggested to cause additional energy dissipation [2]. This can influence post-impact behaviors, such as droplet retraction velocity [6, 7], and may also reduce the net impact force. However, due to the small force magnitude and spatial extent, we believe the effect on impact force and stress distribution in our experimental conditions is secondary.

We have clarified in the main text (Section 2.2) that we did not consider the effect of the wetting ridge in our analysis, and we have added further discussion regarding its possible role in the Supplementary Information.

6. On page 6, Line 234-238, the authors vaguely speculate and does not provide any evidence or analysis to support their claim regarding the observation of central stress peak. Furthermore, the authors appear to misinterpret the findings of Sun et al., who demonstrated that for a droplet impacting an elastic substrate, the stress peak (σ_{zz}) remains non-central, rather than shifting toward the center ($r = 0$). The authors could substantiate their claim by demonstrating that the observations remain consistent across substrates with different elastic moduli? This would help confirm that the result is not merely an experimental artifact.

Thank you for your detailed comment and for pointing out the need for clarity regarding the interpretation of the stress distribution and the observation of central versus non-central peaks. We acknowledge the important findings of Sun et al. (2022), who were the first to experimentally demonstrate the existence of a non-central peak in the normal stress component (σ_{zz}) during droplet impact on elastic substrates. Their observation is qualitatively consistent with theoretical and numerical predictions [8], and we do not intend to contradict their results. At the same time, however, they also mentioned a characteristic that differs from the theoretical prediction for rigid substrates, where shear stress peaks and pressure peaks appear at the same location. Specifically, they found that on elastic substrates, the pressure peak appears behind the shear stress peak, suggesting that this difference may be due to the substrate not being rigid.

In our study, we did not observe a clearly defined non-central peak. Based on this, we offered a hypothesis that the substrate used in our experiment may be softer than that used in Sun et al., which could result in broader deformation and cause the non-central peak to merge with the central stress field—making it indistinguishable. To test this

Figure 3: **Stress distribution at $z = 0$.** Comparison between the theoretical and the measured stress fields. The theoretical stress distribution is calculated using Eq. (2) in the Supplementary Information proposed by Philippi et al. [8]. The experimental data are taken at $V \simeq 1.1$ m/s for 10^0 cSt silicone oil droplets. Dashed line indicates $tV/R = 0.34$, which is the typical time when the maximum impact force occurs for a low-viscosity droplet impacting on a rigid substrate [9, 10].

hypothesis, we conducted additional experiments using substrates with different elastic moduli, and in both cases, the non-central peak remained unclear, as shown in the figure below (also Fig. 4 of the Supplementary Information). This suggests that the absence of a distinct non-central peak is not limited to a particular material or condition, and may instead reflect a consistent feature of impacts on softer substrates.

At the same time, we fully recognize that the reconstruction of σ_{zz} using axisymmetric Abel inversion may suffer from numerical artifacts near the symmetry axis ($r = 0$), due to the amplification of noise in the outer regions. This limitation could artificially enhance the apparent central peak or obscure adjacent local minima. We therefore cannot definitively exclude the possibility that the central peak observed in our data is affected by such reconstruction limitations.

In summary, while our current observations do not show a clear non-central peak as reported by Sun et al., we have verified the reproducibility of this result across different elastic moduli, suggesting it may reflect a physical behavior specific to softer substrates. However, given the possibility of reconstruction artifacts, we present our interpretation with appropriate caution. We have revised the manuscript (Section 2.2) to explicitly clarify this point and added a more detailed discussion in the Supplementary Information.

7. It has been reported in literature that a droplet impacting a soft surface can entrap more air compared to a rigid flat surface, with the size of the entrapped air decreasing based on impact velocity and droplet viscosity. Could the authors comment on whether variations in the air layer thickness beneath the droplet, influenced by different viscosities and substrate softness, affect the stress field distribution? It would be a much stronger story if the elastic modulus of the substrate were varied as well.

Thank you for your important comment regarding air entrapment. The entrapment of air (also known as air cushioning) with a thickness of approximately 0.1 to 10 μm that occurs at the bottom of droplets upon impact with a solid surface delays the contact between the droplet and the substrate, affecting the pressure distribution during the very initial stage of impact (several nanoseconds to several microseconds), particularly reducing water-hammer pressure [11]. The thickness of the air layer generated by air cushioning has been reported to decrease with increasing droplet viscosity [12]. On the other hand, as you pointed out, the thickness of the air layer in air cushioning increases with decreasing substrate elasticity, as reported by Langley et al. [13]. This air entrapment phenomenon ends very early after impact (when the droplet adheres to the substrate), so its influence is considered to be small at the timescale of the maximum impact force of interest in this study (on the order of hundreds of microseconds). However, it may influence the scaling laws of time-dependent changes in impact force or the non-central peak of the stress field. As you pointed out, investigating the effects of substrate elasticity and droplet viscosity on the air cushioning phenomenon beneath the droplet, as well as their influence on impact forces and stress fields, is an important research topic that could contribute to a better understanding of dynamic fluid-structure interactions. By combining a stress measurement system with higher spatio-temporal resolution and high-speed interferometric measurement to achieve simultaneous measurement of the stress field and air layer thickness, it is possible to address this challenge. Although this content exceeds the scope of this study, it is an important research topic and has been mentioned at the end of Section 2.1 in the revised manuscript.

8. The authors can include the relevant literature and cite appropriately, specifically, the paper "Understanding the drop impact phenomenon on soft PDMS Substrate" by S. Mangily and "Droplet impacts onto soft solids entrap more air" by Langely that would strengthen the context of their study and provide a more comprehensive background and situate the current work within the broader research landscape.

Thank you for suggesting these relevant papers. We have cited and reviewed the papers you suggested, as well as relevant studies, in the introduction and results sections in the revised manuscript.

We would like to thank the referee for the constructive comments and attention to detail, which from our point of view, have led to further improvement of the manuscript. The text color of the changes are shown in red in the marked-up manuscript.

Reviewer #3

Summary:

The authors of the manuscript conducted an investigation on the impact forces of liquid droplets on elastic solids, wherein a transition from the inertial regime to the Hertzian regime has been observed. Specifically, they employed the high speed photoelastic tomography to measure the stress on an elastic substrate which was imparted by impacting droplets of different velocities and viscosity. Though the manuscript is well written and organized, it does not meet the requirements of the journal due to the following issues.

Thank you for your valuable feedback on our manuscript. We have carefully considered your comments and have made revisions to address each of your concerns, as outlined below.

1. The impact forces of liquid droplets on solid surfaces have been extensively investigated over the past few years (Soft matter 10 27, 4929-4934 2014; Physical Review Letters 12910, 104501 2022; Phys. Fluids 35, 052104 2023; Physics of Fluids 35, 112111 2023; Phys. Rev. Fluids 8, 113601 2023). However, relevant key findings were not adequately discussed and cited. Moreover, the topic of this study is too specific to attract broad readerships of Nature Communications.

Thank you for your thoughtful and important comment. We sincerely appreciate the opportunity to clarify the positioning and broader relevance of our study. We fully acknowledge that our original manuscript did not sufficiently cite several key contributions in the recent literature, particularly those related to impact forces on rigid substrates. In response, we have revised the Introduction to more clearly position our work within the existing body of research and have appropriately cited the relevant studies.

While it is true that the dynamics of droplet impact forces have been extensively investigated—especially on rigid and superhydrophobic substrates—most previous studies have focused on either droplet morphology (e.g., contact time, spreading) or impact forces without considering substrate deformability. As noted in Cheng et al. (*Annu. Rev. Fluid Mech.*, 2022), our understanding of how substrate mechanics influence force generation remains limited, especially in the context of soft, deformable substrates.

Our study goes beyond prior investigations by addressing this gap and *fundamentally redefining how substrate mechanics alter droplet impact forces*. Specifically, we experimentally reveal a universal *scaling crossover* in maximum impact force, from inertial scaling (typical of rigid substrates) to Hertzian scaling (characteristic of rigid spheres on elastic substrates), governed by a newly identified similarity parameter. This scaling transition captures the combined effects of droplet viscosity, inertia, and substrate elasticity within a unified framework. To reinforce the generality of our findings, we conducted additional experiments using multiple substrates with different elastic moduli (Gel I–III). These confirm that the crossover is not limited to a specific material or viscosity regime, but instead represents a broader physical principle applicable across systems. This universality allows us to provide both *fundamental physical insight* and a *predictive framework* for force generation in droplet impacts on soft materials.

We believe that this perspective broadens the scope of impact physics and provides relevance to fields such as soft matter, fluid–structure interaction, impact mechanics, and biomedical applications. The revised manuscript now reflects this broader context more explicitly.

2. The high speed photoelastic tomography needs to be validated by with other sophisticated force measurement method or numerical simulations. As simple comparison between the results of this work and that of Soto et al. (Soft matter 10 27, 4929-4934 2014), the measured maximum impact forces by the authors are much smaller.

Thank you for your comment. We appreciate the opportunity to clarify the validity of the high-speed photoelastic tomography technique employed in our study. This method enables dynamic quantification of stress fields in soft materials under axisymmetric conditions and has been validated in our previous studies [14, 15].

In those works, we compared reconstructed stress fields from experimental measurements with analytical solutions for static Hertzian contact and confirmed excellent quantitative agreement. In the present manuscript, we further validated the reconstruction by performing additional static contact experiments using the same gel materials. The comparison between the reconstructed stress field and Hertzian theory is shown in Fig. 4(c) of the revised manuscript, again demonstrating strong agreement. These results confirm the reliability of our tomography method in accurately measuring internal stress fields within elastic substrates. We have added this information to the Method section.

Regarding the comparison with Soto et al. [16], we agree that the maximum impact force reported in our study (approximately 15 mN for a

10^1 cSt silicone oil droplet with radius 1.27 mm at 3 m/s) is smaller than the force reported in their study (approximately 45 mN for a water droplet of radius 1.5 mm at similar velocity). We believe this discrepancy arises primarily from the treatment of the z -axis in our reconstruction.

Due to optical reflections and surface curvature near the droplet–substrate interface, the photoelastic signal in the immediate vicinity of the interface is subject to localized distortion. To ensure consistent and artifact-free analysis, we defined $z = 0$ as the position 700 μm below the interface, where the stress field remains axisymmetric and free from optical interference. This approach allows robust and reproducible measurement of the stress distribution while slightly underestimating the absolute peak force.

Importantly, this consistent methodology was applied to all experimental conditions. Thus, although the absolute values of the maximum impact force are reduced, the relative trends and the force scaling behavior—which are the focus of this study—remain valid and unaffected.

We have clarified this point in the revised manuscript to avoid misunderstanding and to support the robustness of our scaling analysis.

3. Only one type of elastic substrates ($E = 47.4$ kPa) has been used in this study, which is thus insufficient to draw the general conclusion as the authored did.

Thank you for your comment. In response to your suggestion, we conducted additional experiments using substrates with different elastic moduli (denoted as Gel II and Gel III) in addition to the elastic substrate used in the original manuscript (denoted as Gel I). Additionally, we reanalyzed the experimental data of Gel I for consistency. Table 1 in the revised manuscript summarizes the elastic moduli E of each gel along with their stress-optic coefficients C , which are used for stress reconstruction. Additionally, we have added the methods for determining E and C for each substrate to the Supplementary Information (Sec. 1). The results of the additional experiments confirmed that, in the scaling laws of Π and Ca , the transition from inertial force scaling to Hertzian impact scaling occurs with increasing droplet viscosity, even when using substrates with different elastic moduli. To quantitatively illustrate the changes in scaling laws, we added the scaling law prefactor and index of $\Pi = BCa^\beta$ for each substrate elasticity-droplet viscosity combination to the Supplementary Information (Sec. 7). Additionally, in the graph of H vs Z (Fig. 3 in the revised manuscript), the crossover phenomenon and data collapse due to the similarity parameter Z were still observed. In particular, in experiments using Gel III with lower elasticity, silicon oil droplets of 10^4 cSt exhibited behavior close to Z^0 ,

Figure 4: **Stress field measurement using high-speed photoelastic tomography.** (a) Schematic of the experimental setup for measuring the stress field in an elastic substrate during a sphere or droplet impact. (b) A typical image of the photoelastic parameters (retardation Δ and orientation ϕ) during the droplet impact. The color indicates Δ and the white arrows indicate ϕ . The length of white arrows correspond to the magnitude of retardation. Comparison of the axial stress within the substrate, Gel I (see Table 1 in the revised manuscript), induced by the contact of the styrol sphere obtained from Hertzian contact theory [17, 18] (left) and from photoelastic tomography (right). The diameter of sphere and contact force are 14.7 mm and 0.55 N, respectively.

indicating behavior closer to the sphere case. This demonstrates that the scaling framework proposed in this study is effective even for different substrate elasticities, thereby strengthening the generality of the conclusion. The results of the additional experiments have been incorporated into the figures in the manuscript, and discussions based on these results have been added.

4. Previous studies have demonstrated that hydrodynamic forces applied on solid surfaces are highly dependent on the drop behaviors (Phys. Fluids 35, 052104 2023; Physics of Fluids 35, 112111 2023; Phys. Rev. Fluids 8, 113601 2023). However, the effects of droplet dynamics on the impact forces were not analyzed at all.

Thank you for your comment. The behavior of liquid droplets is important in determining the shape and magnitude of impact forces. For example, at the moment of maximum impact force ($t \sim 0.2$ ms), low-viscosity droplets spread into a hemispherical shape and exhibit splashing at high-impact velocities, while the spreading of high-viscosity droplets is suppressed by viscous resistance and maintains a shape closer to a sphere. These differences in droplet shape are reflected in the magnitude of stress and impact forces. The paper you suggested focuses mainly on the bounce phenomenon of droplets on superhydrophobic substrates and the second peak of force associated with it. However, even in those studies, it is suggested that the wettability of the substrate has little effect on the first peak of force (the maximum impact force focused on in this study). In the experiments conducted in this study, the droplet-bouncing phenomenon observed in the cited papers did not occur. Therefore, the second peak also did not appear. In this study, we focused on the first peak and investigated how its scaling law is governed by droplet viscosity, velocity, and substrate elasticity. In the revised manuscript, we have appropriately cited the papers you mentioned and explicitly referenced the relationship between droplet dynamics (particularly the initial deformation and spreading process) and droplet shape at the time of maximum impact force in section 2.1 in the revised manuscript.

5. Early studies have explicitly demonstrated that there is a liquid hammer pressure (e.g., Phys. Med. Biol. 36, 1475 1991; J. Fluid Mech. 490, 1 2003) up impacting on solid surfaces. This impact force is much larger than the hydrodynamic force investigated here, and it is much more important in practical applications, e.g., causing the well-known raindrop erosion. Therefore, the authors should include this force term in their studies.

Thank you for pointing this out. The order of magnitude of water hammer pressure is given by ρVc (where c is the speed of sound within

Figure 5: **Effect of water-hammer pressure** Comparison between maximum impact force F_{\max} measured in this study and the force generated by water-hammer pressure via Eq. (3) in the Supplementary Information.

the droplet), with an action time of RV/c^2 and a typical contact area of $(RV/c)^2$ [19, 20, 16, 9]. Therefore, the order of magnitude of the force generated by water hammer pressure can be estimated as $F_w = \rho R^2 V^3 / c$. The plot of the ratio of the maximum impact force F_{\max} measured in this study to F_w is shown in the figure below (also Fig. 5 in Supplementary Information) (here, the sound speed in silicone oil is set to $c = 1000$ m/s). Even in the case where F_{\max}/F_w is smallest, approximately 600, it is clear that F_{\max} is sufficiently larger than F_w in this study. Therefore, this effect is not included in the scaling of F_{\max} in this study. Additionally, it has been pointed out that the air cushioning phenomenon occurring beneath the droplet may mitigate the effects of water hammer pressure [11]. Furthermore, [13] reported that when using a softer substrate, the air layer beneath the droplet becomes larger in the air cushioning phenomenon. Therefore, it is predicted that F_w will be further reduced in the elastic substrate studied in this research. This point has been supplemented in section 2.1 of the manuscript.

References

- [1] A. Carré, J.-C. Gastel, M. E. R. Shanahan, Viscoelastic effects in the spreading of liquids, *Nature* 379 (6564) (1996) 432–434. [doi:10.1038/379432a0](https://doi.org/10.1038/379432a0).
- [2] M. E. R. Shanahan, A. Carre, Viscoelastic Dissipation in Wetting and Adhesion Phenomena, *Langmuir* 11 (4) (1995) 1396–1402. [doi:10.1021/la00004a055](https://doi.org/10.1021/la00004a055).
- [3] N. Xue, L. A. Wilen, R. W. Style, E. R. Dufresne, Droplets sliding on soft solids shed elastocapillary rails, *Soft Matter* 21 (2) (2025) 209–215. [doi:10.1039/D4SM01041H](https://doi.org/10.1039/D4SM01041H).
- [4] L. Hauer, Z. Cai, A. Skabeev, D. Vollmer, J. T. Pham, Phase Separation in Wetting Ridges of Sliding Drops on Soft and Swollen Surfaces, *Physical Review Letters* 130 (5) (2023) 058205. [doi:10.1103/PhysRevLett.130.058205](https://doi.org/10.1103/PhysRevLett.130.058205).
- [5] Z. Cai, R. G. M. Badr, L. Hauer, K. Chaudhuri, A. Skabeev, F. Schmid, J. T. Pham, Phase separation dynamics in wetting ridges of polymer surfaces swollen with oils of different viscosities, *Soft Matter* 20 (36) (2024) 7300–7312. [doi:10.1039/D4SM00576G](https://doi.org/10.1039/D4SM00576G).
- [6] S. Mangili, C. Antonini, M. Marengo, A. Amirfazli, Understanding the drop impact phenomenon on soft PDMS substrates, *Soft Matter* 8 (39) (2012) 10045–10054. [doi:10.1039/C2SM26049B](https://doi.org/10.1039/C2SM26049B).
- [7] A. Alizadeh, V. Bahadur, W. Shang, Y. Zhu, D. Buckley, A. Dhinojwala, M. Sohal, Influence of substrate elasticity on droplet impact dynamics, *Langmuir* 29 (14) (2013) 4520–4524. [doi:10.1021/la304767t](https://doi.org/10.1021/la304767t).
- [8] J. Philippi, P.-Y. Lagrée, A. Antkowiak, Drop impact on a solid surface: Short-time self-similarity, *Journal of Fluid Mechanics* 795 (2016) 96–135. [doi:10.1017/jfm.2016.142](https://doi.org/10.1017/jfm.2016.142).
- [9] X. Cheng, T.-P. Sun, L. Gordillo, Drop impact dynamics: Impact force and stress distributions, *Annual Review of Fluid Mechanics* 54 (1) (2022) null. [doi:10.1146/annurev-fluid-030321-103941](https://doi.org/10.1146/annurev-fluid-030321-103941).
- [10] L. Gordillo, T.-P. Sun, X. Cheng, Dynamics of drop impact on solid surfaces: Evolution of impact force and self-similar spreading, *Journal of Fluid Mechanics* 840 (2018) 190–214. [doi:10.1017/jfm.2017.901](https://doi.org/10.1017/jfm.2017.901).
- [11] T. H. Hoksbergen, R. Akkerman, I. Baran, Liquid droplet impact pressure on (elastic) solids for prediction of rain erosion loads on wind turbine blades, *Journal of Wind Engineering and Industrial Aerodynamics* 233 (2023) 105319. [doi:10.1016/j.jweia.2023.105319](https://doi.org/10.1016/j.jweia.2023.105319).

- [12] K. Langley, E. Q. Li, S. T. Thoroddsen, Impact of ultra-viscous drops: Air-film gliding and extreme wetting, *Journal of Fluid Mechanics* 813 (2017) 647–666. doi:[10.1017/jfm.2016.840](https://doi.org/10.1017/jfm.2016.840).
- [13] K. R. Langley, A. A. Castrejón-Pita, S. T. Thoroddsen, Droplet impacts onto soft solids entrap more air, *Soft Matter* 16 (24) (2020) 5702–5710. doi:[10.1039/D0SM00713G](https://doi.org/10.1039/D0SM00713G).
- [14] Y. Yokoyama, B. R. Mitchell, A. Nassiri, B. L. Kinsey, Y. P. Korkolis, Y. Tagawa, Integrated photoelasticity in a soft material: Phase retardation, azimuthal angle, and stress-optic coefficient, *Optics and Lasers in Engineering* 161 (2023) 107335. doi:[10.1016/j.optlaseng.2022.107335](https://doi.org/10.1016/j.optlaseng.2022.107335).
- [15] Y. Yokoyama, S. Ichihara, Y. Tagawa, High-speed photoelastic tomography for axisymmetric stress fields in a soft material: Temporal evolution of all stress components, *Optics and Lasers in Engineering* 178 (2024) 108224. doi:[10.1016/j.optlaseng.2024.108224](https://doi.org/10.1016/j.optlaseng.2024.108224).
- [16] D. Soto, A. B. D. Larivière, X. Boutillon, C. Clanet, D. Quéré, The force of impacting rain, *Soft Matter* 10 (27) (2014) 4929–4934. doi:[10.1039/C4SM00513A](https://doi.org/10.1039/C4SM00513A).
- [17] K. L. Johnson, *Contact Mechanics*, Cambridge University Press, Cambridge, 1985. doi:[10.1017/CB09781139171731](https://doi.org/10.1017/CB09781139171731).
- [18] B. Mitchell, Y. Yokoyama, A. Nassiri, Y. Tagawa, Y. P. Korkolis, B. L. Kinsey, An investigation of hertzian contact in soft materials using photoelastic tomography, *Journal of the Mechanics and Physics of Solids* 171 (2023) 105164. doi:[10.1016/j.jmps.2022.105164](https://doi.org/10.1016/j.jmps.2022.105164).
- [19] J. E. Field, The physics of liquid impact, shock wave interactions with cavities, and the implications to shock wave lithotripsy, *Physics in Medicine & Biology* 36 (11) (1991) 1475. doi:[10.1088/0031-9155/36/11/007](https://doi.org/10.1088/0031-9155/36/11/007).
- [20] M. A. Nearing, J. M. Bradford, R. D. Holtz, Measurement of force vs. Time relations for waterdrop impact, *Soil Science Society of America Journal* 50 (6) (1986) 1532–1536. doi:[10.2136/sssaj1986.03615995005000060030x](https://doi.org/10.2136/sssaj1986.03615995005000060030x).

We would like to thank the referee for the constructive comments and attention to detail, which from our point of view, have led to further improvement of the manuscript. The text color of the changes are shown in red in the marked-up manuscript.

Reviewer #3

Summary:

From the rebuttal one can read that the authors have managed to improve the manuscript following the suggestions of three reviewers. However, some key issues need to be further addressed before considering for publication.

Thank you for your valuable feedback on our manuscript. We have carefully considered your comments and have made revisions to address each of your concerns, as outlined below.

1. All three reviewers have suggested to perform additional measurements on other soft substrates to demonstrate the robustness of the finding. Though new experimental results were added in the revised manuscript, they were obtained from solid substrates with comparable moduli ($E=44$ kPa and 17.9 kPa) as that in the previous version ($E=47.4$ kPa). Instead, the authors should conduct experiments on solid substrates with modulus difference of orders of magnitude. From Fig. 3a, it can be found that the experimental data already exhibits strong scattering, which suggests the importance of conducting experiments on a wide range of solid substrates.

Thank you for your thoughtful and important comment. In response, we performed additional measurements on substrates whose elastic moduli differ by orders of magnitude and incorporated the results into the revised manuscript. Specifically, beyond the gels in the previous versions ($E = 47.4, 44,$ and 17.9 kPa), we newly prepared a stiffer gels, up to $E \approx 126$ kPa, and a much softer gelatin-based gel, $E \approx 2.7$ kPa. Figure 1 shows the results of the measurements for the elastic moduli and stress-optic coefficients of the gels.

Across all substrates, we collected more than 500 individual impact-force/stress data points in total. We also re-labelled the gels as Gel I–VI in descending order of E for consistent cross-referencing in text and figures. Because the 2.7 kPa gelatin substrate (Gel VI) is extremely soft and fragile, repeated runs were not feasible. We therefore report one experiment per fall height (8, 10, 15, 20, 30 cm) at $\nu = 1$ cSt silicone oil for that substrate. Following best practice, we do not plot error bars for this particular data set and clearly note this limitation in the main text of the revised manuscript. With these additions, we updated the scaling plots (Π vs. Ca and H vs Z) in Figure 2 and highlighted the

Figure 1: **Physical properties of the elastic substrates** (a) Loading force F acting on the elastic substrate measured by the electric balance against the surface indentation δ . The dashed lines are calculated by Eq. (1) in the Supplementary Information using the averaged elastic modulus. (b) E against F for different substrates. The dashed lines indicate the average value of each. (c) The stress-optic coefficient C with different loading forces F . The dashed lines indicate the average value of each.

newly acquired data while rendering the earlier data semi-transparent for clarity. Across the extended set, the data continue to exhibit a continuous crossover between the inertial and Hertzian scalings.

Notably, the new measurements reinforce our main conclusion. On the softest substrate (Gel VI, $E \simeq 2.7$ kPa), even for the low-viscosity case ($\nu = 1$ cSt) the Π - Ca slope shifts toward the Hertzian trend (3/5), consistent with the view that decreasing substrate elasticity promotes Hertz-like response. The reconstructed stress fields showed a pronounced tensile (negative) stress region near the liquid–solid contact line (see Fig. 3(a)). This feature is also present the rigid sphere impact case and high-viscous droplet (10^4 cSt silicone oil) case (see Fig. 1 of the main text and Fig. 2 of the Supplementary Information). Correspondingly, the temporal evolution of impact force is more symmetric about the time of the maximum force (Fig. 3(b)), consistent with a trend toward Hertzian impact behavior.

To our knowledge, our experiment represents the softest substrate on which droplet impact stress fields have been measured to date. Achieving further reducing the substrate elasticity to get more systematic experiment will require gels that are both very soft and sufficiently photoelastic, while remaining practical for droplet-impact experiments. Developing such materials is technically challenging and, in itself, a worthwhile topic for future research. We replaced the scaling plot and described the discussions on it in the revised manuscript.

2. Compared to existing results of measuring impact forces of liquid droplets, the authors have ascribed the absence of a second peak value to that no bouncing

Figure 2: **Self-similar structure of the maximum impact force.** (a) Π vs. Ca . The gray square markers represent the data of the sphere case. The inset of (a) represents Π vs. θ . The red solid line indicates Hertzian impact scaling, Eq. (2) in the main text, and the blue solid line indicates inertial force scaling, Eq. (1) in the main text. (b) H vs. $Z = \theta^\alpha Ca$ with $\alpha = 0.24$. The red solid line indicates Eq. (3) in the main text and the blue solid line indicates Eq. (4) in the main text. The inset of (b) shows I vs. α . The error bars represent one standard deviation over three impacts.

phenomenon occurred in their experiments. However, the fact is that the second peak force in droplet impact is produced by the cavity collapse (Physical Review Letters 12910, 104501 2022; Phys. Fluids 35, 052104 2023; Physics of Fluids 35, 112111 2023; Phys. Rev. Fluids 8, 113601 2023), which almost happens in all impact events regardless of the viscosity of the liquid and wetting property of the solid substrate (Annual Review of Fluid Mechanics 48, 365–391 2016; Physical Review Letters 96, 124501 2006). This inconsistency should be carefully explained as it relates to reliability of the force/stress measurement methodology.

Thank you for the helpful clarification. First, in the same fluid family and comparable impact condition (high Weber number $We = \rho V^2 R/\gamma \simeq 100 \sim 500$) as our study (see Fig. 4), Gordillo et al. reported single-peaked force histories for silicone-oil drops [1]. A clear primary peak at impact, followed by a monotonic decay with no second peak, was observed under their conditions. Our silicone-oil experiments yield the same single-peaked profile, consistent with their observations. Second, the second force peak is pronounced on non-wetting substrates when the droplet exhibits the cavity collapse inside the droplet following the retraction behavior, as reported previously [2, 3, 4, 5]. Zhang et al. showed an explicit dependence on contact angle [4]. For sufficiently high wettability (e.g., contact angles below $\sim 40^\circ$), the second peak can

Figure 3: **Representative reconstructed stress fields and force profile.** (a) Spatio-temporal distribution of the stress field (bottom-left panels) photoelastic parameters, Δ and ϕ , and (bottom-right panels) reconstructed axial stress σ_{zz} when a droplet impact on the elastic substrate Gel VI ($E = 2.7$ kPa) with $V \simeq 2.37$ m/s. In the bottom-left panels, the colormap indicates the retardation Δ and white arrows indicate the orientation ϕ . The arrow length corresponds to the magnitude of retardation shown in the colorbars. The gray-scale images of the impactor on the upper panel are obtained by subtracting the background image without the impactor. (b) Temporal evolution of the impact force with different impact velocities during a 1 cSt silicone oil droplet impact on the elastic substrate Gel VI ($E = 2.7$ kPa).

be strongly attenuated or absent. Third, the cavity collapse follows the retraction phase after maximum spread of the droplet [6, 7]. In our measurements on substrates wettable to silicone oil, we did not observe the retraction dynamics that would lead to a cavity, nor any subsequent cavity collapse. Therefore, the absence of a second peak in our force records reflects that this phenomenon did not occur under our conditions, rather than any limitation of the force/stress measurement methodology.

We also wish to correct our earlier wording. As you pointed out, our previous response did not describe the physical origin of the second peak with sufficient precision, and attributing it to droplet bouncing was inaccurate. In the revised manuscript, we now state explicitly that, when it occurs, the second peak originates from cavity collapse, and we have added the representative references cited.

To our knowledge, direct, time-resolved measurements of the substrate stress field precisely at cavity collapse remain outstanding; exploring possible roles of substrate elasticity in that regime is a valuable direction for future work. We have added these points to the revised manuscript.

Figure 4: **Experimental parameter space explored in this study.** Experimental conditions explored in this study plotted on the Ohnesorge and Weber numbers plane. The axes are $Oh = \eta / \sqrt{\rho R \gamma}$ and $We = \rho V^2 R / \gamma$.

3. The effect of air cushion on the stress distribution has been discussed; however, no explicit comments/conclusions were given/made. The appearance of an air cushion below impacting liquid droplets is a ubiquitous phenomenon, and it can even happen on very rigid solid substrates, making droplets bouncing off (Nature Physics 11, 48–53 2015). Therefore, it is very crucial to clarify whether it affects the impact stress applied on the solid substrates. Actually, performing a comparative experiments of droplet impact in air and in vacuum chamber can provide some clues. Regarding the second issue, does the absence of the second peak force result from the air cushion?

We appreciate the reviewer’s comment regarding the role of the air cushion in drop impact. To assess its relevance in our regime, we attempted AFM measurements of the gel surface. Although the soft gel could not be reliably scanned, the acrylic mold used for gel fabrication showed a surface roughness of 10–100 nm (Fig. 5). The gel surface is expected to be slightly rougher than it, which is comparable to the characteristic early-time air-film thickness ($\sim 0.1 \mu\text{m}$) [8, 9, 10, 11]. Prior studies show that the air film is micron or sub-micron in thickness and collapses within microseconds at moderate–high Weber numbers. Given the similar scales, the air layer is expected to rupture and direct liquid–solid contact forms before $\sim 0(10^2) \mu\text{s}$, when our peak force occurs. Thus, air cushioning is unlikely to significantly influence our measured forces or stresses. We have clarified this point in the manuscript.

Figure 5: **Surface characterization of the acrylic mold.** Atomic force microscopy (AFM) measurement of the surface of the acrylic container used as the mold for gel preparation.

For smoother substrates, the air layer can persist and redistribute pressure, e.g., screening water-hammer loading and shifting the pressure maximum off-axis [12, 13, 14]. As emphasized in [14], systematic stress measurements under varied surface roughness and ambient pressure remain limited. We agree that reduced-pressure experiments would be decisive to isolate air-cushion effects and now note this as future work. Finally, because the air layer is assumed to be disrupted early under our conditions, we do not attribute the absence of a second force peak to air cushioning.

References

- [1] L. Gordillo, T.-P. Sun, X. Cheng, Dynamics of drop impact on solid surfaces: Evolution of impact force and self-similar spreading, *Journal of Fluid Mechanics* 840 (2018) 190–214. [doi:10.1017/jfm.2017.901](https://doi.org/10.1017/jfm.2017.901).
- [2] B. Zhang, V. Sanjay, S. Shi, Y. Zhao, C. Lv, X.-Q. Feng, D. Lohse, Impact forces of water drops falling on superhydrophobic surfaces, *Physical Review Letters* 129 (10) (2022) 104501. [doi:10.1103/PhysRevLett.129.104501](https://doi.org/10.1103/PhysRevLett.129.104501).
- [3] J. Li, A. Oron, Y. Jiang, Droplet jump-off force on a superhydrophobic surface, *Physical Review Fluids* 8 (11) (2023) 113601. [doi:10.1103/PhysRevFluids.8.113601](https://doi.org/10.1103/PhysRevFluids.8.113601).
- [4] B. Zhang, C. Ma, H. Zhao, Y. Zhao, P. Hao, X.-Q. Feng, C. Lv, Effect of wettability on the impact force of water drops falling on flat solid surfaces, *Physics of Fluids* 35 (11) (2023) 112111. [doi:10.1063/5.0173851](https://doi.org/10.1063/5.0173851).
- [5] B. Zhang, H. Zhao, Y. Zhao, P. Hao, C. Lv, Impact force of ring bouncing on superhydrophobic surface with a bead, *Physics of Fluids* 35 (5) (2023) 052104. [doi:10.1063/5.0152170](https://doi.org/10.1063/5.0152170).
- [6] D. Bartolo, C. Josserand, D. Bonn, Singular Jets and Bubbles in Drop Impact, *Physical Review Letters* 96 (12) (2006) 124501. [doi:10.1103/PhysRevLett.96.124501](https://doi.org/10.1103/PhysRevLett.96.124501).
- [7] C. Josserand, S. Thoroddsen, Drop impact on a solid surface, *Annual Review of Fluid Mechanics* 48 (1) (2016) 365–391. [doi:10.1146/annurev-fluid-122414-034401](https://doi.org/10.1146/annurev-fluid-122414-034401).
- [8] M. Mani, S. Mandre, M. P. Brenner, Events before droplet splashing on a solid surface, *Journal of Fluid Mechanics* 647 (2010) 163–185. [doi:10.1017/S0022112009993594](https://doi.org/10.1017/S0022112009993594).
- [9] S. Mandre, M. P. Brenner, The mechanism of a splash on a dry solid surface, *Journal of Fluid Mechanics* 690 (2012) 148–172. [doi:10.1017/jfm.2011.415](https://doi.org/10.1017/jfm.2011.415).
- [10] P. Chantelot, D. Lohse, Drop impact on superheated surfaces: From capillary dominance to nonlinear advection dominance, *Journal of Fluid Mechanics* 963 (2023) A2. [doi:10.1017/jfm.2023.290](https://doi.org/10.1017/jfm.2023.290).
- [11] J. de Ruiter, Dynamics of Collapse of Air Films in Drop Impact, *Physical Review Letters* 108 (7) (2012). [doi:10.1103/PhysRevLett.108.074505](https://doi.org/10.1103/PhysRevLett.108.074505).
- [12] T. H. Hoksbergen, R. Akkerman, I. Baran, Liquid droplet impact pressure on (elastic) solids for prediction of rain erosion loads on wind turbine blades, *Journal of Wind Engineering and Industrial Aerodynamics* 233 (2023) 105319. [doi:10.1016/j.jweia.2023.105319](https://doi.org/10.1016/j.jweia.2023.105319).

- [13] T. H. Hoksbergen, R. Akkerman, I. Baran, Rain droplet impact stress analysis for leading edge protection coating systems for wind turbine blades, *Renewable Energy* 218 (2023) 119328. [doi:10.1016/j.renene.2023.119328](https://doi.org/10.1016/j.renene.2023.119328).
- [14] X. Cheng, T.-P. Sun, L. Gordillo, Drop impact dynamics: Impact force and stress distributions, *Annual Review of Fluid Mechanics* 54 (1) (2022) null. [doi:10.1146/annurev-fluid-030321-103941](https://doi.org/10.1146/annurev-fluid-030321-103941).

We would like to thank all reviewers for their time and constructive feedback throughout the review process. We appreciate their helpful comments in the earlier rounds and are grateful for the final positive assessment indicating that the manuscript is suitable for publication.